# Immersive Technologies Targeting Spatial Memory Decline: A Systematic Review

**DOI:** 10.3390/biomedicines13092105

**Published:** 2025-08-29

**Authors:** Lucía Solares, Sara García-Navarra, Tania Llana, Sara Garces-Arilla, Marta Mendez

**Affiliations:** 1Department of Psychology, Faculty of Psychology, University of Oviedo, Plaza Feijoo s/n, 33003 Oviedo, Spain; uo271781@uniovi.es (L.S.); uo302771@uniovi.es (S.G.-N.); llanatania@uniovi.es (T.L.); 2Instituto de Neurociencias del Principado de Asturias (INEUROPA), Plaza Feijoo s/n, 33003 Oviedo, Spain; 3Instituto de Investigación Sanitaria del Principado de Asturias (ISPA), Av. del Hospital Universitario s/n, 33011 Oviedo, Spain; 4Department of Psychology and Sociology, Faculty of Social and Human Sciences, IIS Aragón, University of Zaragoza, 44003 Teruel, Spain; sgarces@unizar.es; 5BioMedical Engineering Center (BME), University of Oviedo, Campus Viesques, 33204 Gijón, Spain

**Keywords:** spatial memory, virtual reality, mixed reality, aging, navigation, immersive technology, neuropsychology, cognitive assessment

## Abstract

**Background/Objectives**: The ability to preserve cognitive health in aging populations increasingly relies on early detection and intervention in neurodegenerative processes. Spatial memory, a fundamental cognitive ability supporting navigation, environmental awareness, and daily independence, often deteriorates in the preclinical stages of neurodegenerative diseases. However, conventional assessment tools frequently lack ecological validity and fail to capture the multifaceted nature of spatial cognition in real-world contexts. This systematic review aims to examine the application of immersive technologies, specifically Immersive Virtual Reality (VR) and Mixed Reality (MR), in the evaluation and rehabilitation of spatial memory. **Methods**: Following PRISMA guidelines, a total of 42 peer-reviewed studies were selected from SCOPUS, Web of Science, and PubMed databases. We included original, peer-reviewed studies that assessed spatial memory or cognition using VR/MR in adults aged ≥50 or clinical populations at neurodegenerative risk and reported quantitative data or diagnostic validity. A narrative synthesis was performed to examine the most employed immersive tools, assessing their benefits, limitations, and practical challenges. **Results**: Findings indicate substantial variability in diagnostic sensitivity, ecological validity, and user engagement across platforms. Nevertheless, the evidence supports the potential of immersive environments as effective tools for the early detection of spatial disorientation and cognitive decline, particularly in at-risk populations such as individuals with Mild Cognitive Impairment and Alzheimer’s Disease. **Conclusions**: Immersive and semi-immersive VR technologies represent a promising advancement in spatial memory assessment and rehabilitation, offering scalable solutions for both clinical and home-based interventions in aging populations.

## 1. Introduction

The global increase in life expectancy has led to a heightened need to identify scalable, accurate, and ecologically valid tools for monitoring cognitive health in aging populations. The overall prevalence of Mild Cognitive Impairment (MCI) among the aging population is 19.7% and the estimated global number of people living with dementia is projected to rise to 152.8 million by 2050 [1,2]. In particular, spatial memory is frequently impaired in the early stages of Alzheimer’s disease (AD) and other neurodegenerative conditions [3]. Although this domain has been less studied than other forms of memory in patients with MCI and AD, its assessment may offer greater diagnostic value than traditional memory tests [4]. 

Spatial memory, defined as the cognitive ability to retain and recall the spatial configuration of one’s surroundings, enables individuals to develop mental representations of the environment that support both object localization and spatial orientation [5]. This function supports navigation, a crucial skill that allows subjects to follow routes and recognize landmarks in familiar and novel environments, being integral to a wide range of daily activities, from locating personal items to navigating complex environments [6].

Conventional methods for the evaluation of spatial memory include paper-and-pencil tests, structured laboratory tasks, and real-world navigation tasks. These assessments include the administration of questionnaires, which are designed to assess self-reported spatial memory abilities, the detecting of spatial navigation impairments in real-world environments through the assessment of navigation complaints [7,8], and subjective spatial memory performance [9,10]. In addition, laboratory tasks that frequently utilize maze-based paradigms, initially developed for the study of spatial cognition in animal models, offer structured environments that emulate real-world navigation in laboratory settings [11,12]. Exploration of real-world settings, including urban streets, buildings, and natural environments such as parks, has been extensively utilized for the evaluation of wayfinding and spatial navigation abilities. These environments offer strong ecological validity, providing insights into how people navigate complex, dynamic spaces. Research in this area often focuses on how individuals form cognitive maps through exploration and how well they can later recall routes or recognize environmental landmarks [13]. Structured navigation tasks, such as those found in university campuses or hospitals, are frequently utilized to assess spatial learning. Participants are typically free to explore, followed by recall-based tasks that measure their memory for spatial layouts, key locations, and decision points [14]. All these methods have made significant contributions to our understanding of age-related cognitive changes in spatial memory. However, these methods often suffer from lack of adaptability and logistical constraints that hinder large-scale or personalized implementation [15].

The high prevalence of neurodegenerative diseases, which compromise spatial memory functioning in the aging population, highlights the necessity of developing new diagnostic tools. The integration of advanced technologies, including immersive virtual reality (iVR) and mixed reality (MR), has the potential to enhance ecological validity, facilitate the manipulation of experimental variables, and simplify measurement and data recording [16]. iVR and MR technologies differ from non-immersive virtual reality (niVR) in that iVR and MR employ head-mounted displays (HMDs), while non-immersive VR utilizes screens. In the iVR experience, users are visually immersed in a computer-generated environment, with the real world completely hidden, whereas in MR, computer-generated 3-dimensional images are superimposed onto real-world surfaces [17]. By utilizing HMDs and motion tracking, iVR generates multisensory, interactive environments that closely resemble real-life navigation scenarios [18,19]. These platforms also enable the creation of interactive, lifelike environments where researchers can simulate real-world navigation challenges with high precision, while maintaining experimental control. 

Recent studies have highlighted the potential of immersive technologies, including iVR and MR, to help address the challenges of integrating traditional clinical assessments with real-world cognition, which is consistent with the public health goals of aging in place and maintaining independence. Recent advances in this field have opened new avenues for assessing and training spatial memory in aging populations [20]. Immersive platforms also allow integration of behavioral, physiological, and neuroimaging data, enabling comprehensive investigation of the neural mechanisms underlying spatial memory [21]. Moreover, the combination of immersive technologies with the use of AI-driven diagnostic tools, such as machine learning, could overcome the limitations of traditional tests employed for the early detection of neurodegenerative disease in the elderly [22].

The assessment potential of VR-based tools has become especially relevant in the early detection of mild cognitive impairment (MCI) and AD. A substantial body of research has demonstrated that specific spatial memory tasks, such as path integration and object-location memory, are impaired in the preclinical and prodromal stages of AD and may be more sensitive than traditional tests [20,23]. Furthermore, these tasks engage brain regions, such as the hippocampus and entorhinal cortex, that are amongst the earliest affected by AD pathology [24]. Consequently, immersive spatial assessments have the capacity to function as ecologically valid and diagnostically informative tools for identifying individuals at risk. 

Beyond the scope of assessment, immersive technologies offer novel opportunities for cognitive rehabilitation, providing adaptative and engaging training experiences. VR-based spatial memory training programs have the capacity to be adapted to individual performance, can be repeated without inducing fatigue, and can be implemented in a variety of environments, ranging from clinics to homes. These programs offer proactive navigational assistance using remote monitoring by physicians and caregivers. The incorporation of multisensory feedback (e.g., olfactory, auditory), real-time interaction, and embodied navigation has the potential to further enhance learning and memory consolidation [20]. Nevertheless, questions persist regarding the long-term effectiveness, usability, and generalizability of such interventions in heterogeneous older populations. The implementation of iVR is not without its challenges. These challenges include cybersickness, technical costs, and the necessity for standardization across protocols, which continue to impede the adoption of VR technology on a broad scale [25,26]. 

This systematic review, conducted in accordance with the PRISMA guidelines, aims to evaluate the use of immersive technologies in the assessment and rehabilitation of spatial memory among older adults. A review of recent original studies that utilized iVR or MR platforms to examine spatial memory in aging was conducted. Key outcomes include the types of spatial tasks employed, the technologies used, and their effectiveness in both assessment and intervention contexts. For instance, iVR tasks have shown higher diagnostic sensitivity by detecting MCI more accurately than traditional paper-and-pencil tests, while iVR-based training interventions have demonstrated efficacy by reducing navigation errors in individuals with AD. In this study, we methodically assess the current state of the field, identify knowledge gaps and methodological limitations, and outline directions for future research and public health innovation. A particular emphasis is placed on the potential of immersive technologies to facilitate early diagnosis, continuous monitoring, and targeted interventions aimed at enhancing spatial memory resilience and promoting autonomy in older adults. 

## 2. Materials and Methods

### 2.1. Search Strategy

In line with the Preferred Reporting Items for Systematic Reviews and Meta-Analyses (PRISMA) guidelines [27], this systematic review was carried out to assess original peer-reviewed research articles examining the use of immersive technologies, specifically iVR, MR, and related digital tools, in the assessment or monitoring of spatial memory in aging populations. A comprehensive search of Scopus, Web of Science, and PubMed was conducted for studies published between January 2020 and 27 March 2025, with the objective of capturing recent advances. The search was limited to original research articles and was developed using a combination of terms (MeSH terms) and free-text keywords. The following Boolean combination was applied: (“spatial memory” OR “navigation” OR “wayfinding”) AND (“virtual reality” OR “VR” OR “mixed reality” OR “MR” OR “augmented reality” OR “immersive technology”) AND (“aging” OR “older adults” OR “elderly” OR “healthy aging” OR “cognitive decline” OR “Alzheimer’s disease” OR “mild cognitive impairment”).

### 2.2. Study Selection

The eligibility criteria employed in this systematic review included the following: (1) original research articles, (2) English-language studies due to translation limits, (3) studies assessing the spatial memory or spatial cognition using immersive technologies (VR, MR), (4) studies involving human participants aged 50 years or older, or clinical groups with neurodegenerative risk (mixed-age samples were included only if data form participants ≥50 years or relevant clinical subgroups could be separately extracted), and (5) studies reporting quantitative outcomes related to spatial cognition or diagnostic validity. Studies failing to meet these criteria were excluded, including the following: (1) case reports, (2) animal studies or purely technical simulations without human cognitive data, (3) studies that only assessed general memory or executive functions without spatial components, and (4) studies that only assessed usability. The selection of the studies was conducted independently by two reviewers, with any discrepancies addressed through consultation with a third reviewer. No automation tools were employed during data extraction. The data were extracted using a standardized form that captured sample characteristics, technological modalities, task types, outcome measures, and principal findings. The data extraction process was performed by one researcher independently, with the process subsequently verified by a second researcher. Disagreements were resolved by a third reviewer or consensus-based discussion. 

A total of 338 records were identified from searches in all databases (139 from Scopus, 9 from Web of Science, and 105 from PubMed). Of these, 42 studies were incorporated into the final review. The PRISMA flowchart of the search process is presented in Figure 1. A total of 116 articles were selected for a full-text screening following the removal of 134 duplicated articles and 88 articles excluded by initial screening based on title and abstract. Following the application of the exclusion criteria, 74 studies were excluded from the analysis. Four articles were not conducted in humans, 13 studies did not use immersive technologies, 9 studies used technical simulations, 41 articles did not report spatial memory outcomes, and 7 studies only assessed usability.

This systematic review was registered in the OSF (Registration: https://doi.org/10.17605/OSF.IO/K2UX3). Despite the absence of formal protocol, the methodological approach was delineated prior to the analysis and synthesis of the data. The registration of the protocol, available in the OSF repository, was completed after the data extraction phase had begun. 

### 2.3. Quality Assessment

One reviewer conducted the initial quality assessment of all included studies using standardized critical appraisal checklists from the Joanna Briggs Institute (JBI) [28], which was subsequently verified by a second reviewer. Discrepancies were resolved through discussion, with a third reviewer consulted when consensus was not reached. In line with JBI guidance, methodological quality is reported descriptively through narrative synthesis in the Section 3 (Results), while detailed item-level assessments are provided in the Appendix A. 

## 3. Results

### 3.1. Included Studies

This systematic review synthesized findings from 42 studies published between 2019 and 2025 that investigated spatial memory in participants aged ≥50 or at neurodegenerative risk that used immersive technologies (VR, MR) and reported quantitative spatial memory outcomes. We excluded case reports, animal studies, technical simulations without cognitive data, studies assessing only general cognition, and usability-only tests. 

In terms of technological modalities, most studies (over 70%) employed iVR with HMDs such as the HTC Vive, Oculus Rift, or Meta Quest series. Around a quarter used semi-immersive or desktop-based VR systems, often relying on flat screens or treadmills for navigation. Table 1 presents a comprehensive overview of the sample characteristics, technological modalities employed, task modalities, primary outcome measures, and key findings for each study.

Most of the studies were cross-sectional designs, typically comparing healthy older adults with individuals with MCI, AD, or younger controls. A smaller number implemented intervention-based protocols (e.g., short-term VR training or rehabilitation programs [63,64,65,67,69]) to evaluate improvements in spatial recall and navigation strategies. Only a very limited subset employed longitudinal approaches, often in the context of biomarker follow-up [44] or multisession learning paradigms [62,64]. 

The geographic distribution of the included research was broad, with studies conducted across Europe, North America, and Asia, thereby including at least three continents and over ten countries (for example, the United Kingdom [33,34] and Spain [35] in Europe, the United States [53] in North America, and China [67], Hong Kong [66], and Japan [48,63] in Asia). This global representation serves to enhance the external validity of the findings. However, it should be noted that most of the cohorts consisted of relatively high-functioning and educated participants.

### 3.2. Sample Characteristics of Included Studies

The 42 studies included in this systematic review included a total of more than 3000 participants from a variety of age groups, cognitive profiles, and geographic locations. The majority of studies focused on older adults, with participants typically aged 60 years and older. A significant proportion of studies included participants with MCI or AD. Others included healthy older adults, younger controls, or both, to allow comparative analyses of age-related differences in spatial memory performance [34,36,48,50,52,60]. The mean participant age across studies ranged approximately from 21 to 84 years, with standard deviations typically between 3 and 9 years when reported. Across studies, sample sizes varied widely, from as few as 7 participants in highly focused pilot studies [62,67] to as many as 177 participants in broader cohort-based investigations [48]. 

In terms of cognitive status, approximately 41% of studies included only healthy older adults, 36% included individuals with MCI (including both biomarker-positive and -negative subtypes), 10% included participants with AD, and 14% included those with subjective cognitive decline. Across the total sample, about 38% were cognitively healthy, 21% were diagnosed with MCI, 5% with AD, 4% with subjective cognitive decline, and the remaining were mixed or unspecified.

While few studies achieved gender balanced samples [58,60], many did not report results stratified by gender or socioeconomic background, limiting assessment of population diversity. Additionally, education levels were inconsistently reported. However, among studies that reported education levels, there was a tendency towards participants with higher levels of education, with most MCI or AD participants exhibiting at least a primary or secondary education level [31]. The overrepresentation of educated participants may limit the generalizability of findings to more vulnerable populations with less education. The reporting on ethnicity or cultural backgrounds was scarce. The majority of studies were conducted in Western, educated, industrialized populations. This represents a major gap in representativeness, particularly in light of the expanding prevalence of dementia in non-Western regions. 

Recruitment methods varied across studies. Clinical samples were often drawn from outpatient memory clinics or hospital databases, while control groups were typically recruited from university campuses or community centers. This resulted in occasional imbalances in sample sizes between groups. Only 30% of studies explicitly matched clinical and control groups by age, sex, and education. Attrition and recruitment bias were not systematically reported. Only a few studies addressed dropout due to task disengagement or technical issues [66], while others did not account for incomplete data or low-quality responses in virtual tasks.

Most studies used cognitive screening instruments such as MMSE or MoCA to ensure that participants met specific cognitive inclusion criteria. Clinical classifications of MCI and AD varied, with several studies using standard neuropsychological diagnostic criteria and a few using biomarker-based classifications [31,32,54]. A substantial proportion of studies, over 50%, included a control group, allowing comparisons between groups (e.g., healthy vs. MCI, MCI+ vs. MCI−, or AD vs. young adults). However, some exploratory studies have used single-group designs that focused solely on feasibility, usability, or preliminary cognitive effects [62,63,67].

### 3.3. Technological Platforms and Immersion Levels

Desktop VR typically involves computer-based virtual environments navigated via a keyboard, joystick, or mouse, offering limited sensory engagement. Semi-immersive VR makes use of large projection screens or cave-like environments that increase perceptual immersion while participants remain physically seated or walking on a treadmill. AR overlays digital content onto the real-world environment, blending physical and virtual stimuli. Fully iVR delivered through HMDs enables stereoscopic 3D vision and real-time motion tracking, allowing participants to move naturally within virtual space. These differences have significant implications for the user’s sense of presence, the realism of the spatial experience, and the ecological validity of spatial tasks. 

The revised immersive technologies for spatial memory research ranged from low-immersion desktop VR [36,50,55] to fully immersive iVR HMDs, included in 25 of the reviewed studies [37,47,53,54,61], with semi-immersive systems and AR in between [59]. 

The prevalence of desktop-based and semi-immersive systems can be attributed to their accessibility, cost-effectiveness, and lower technical demands, particularly in research conducted in non-laboratory or clinical settings. These platforms typically involve flat-screen displays and rely on mouse-keyboard or joystick inputs for navigation. Despite their relative lack of full-body interactivity and sensory richness of HMDs, they can still simulate spatial environments with considerable visual realism. Cammisuli et al. [32] integrated a real-world detour navigation test, employing wearable sensors to provide an ecologically valid method for assessing spatial orientation in ambulatory contexts. This design exemplifies the potential application of systems beyond the limitations of laboratory environments, thereby integrating digital assessment with real-life navigation performance. Diersch et al. [40], used a photorealistic desktop-based virtual city to study age-related differences in hippocampal and retrosplenial cortex activity during directional pointing tasks. Their findings showed that older adults displayed reduced spatial learning and altered neural activity compared to younger adults, reinforcing the utility of semi-immersive systems for neurocognitive research. Likewise, Ladyka-Wojcik et al. [49] implemented egocentric and allocentric object-location tasks using desktop VR and found that egocentric-to-allocentric frame switching was particularly difficult for older adults with lower cognitive reserve. Despite their reduced immersion, semi-immersive platforms are often better tolerated by participants with limited mobility, sensory deficits, or limited experience with technology. This renders them a valuable option for more vulnerable subgroups in the field of aging research and public health applications. 

By comparison, AR has been less frequently adopted in this field. In our review, only one study has explored the potential of AR to enhance spatial memory tasks. This technology overlay digital content onto the physical environment, thereby facilitating real-world navigation augmented by cognitive cues or feedback. AR systems have demonstrated particular efficacy in ecologically valid training or rehabilitation applications, as they blend real and virtual stimuli without isolating users from their surroundings. Qiu et al. [59] used the Microsoft HoloLens 2 smartglasses to conduct indoor landmark-based wayfinding tasks. Participants in the AR condition showed better spatial performance, including faster task completion and more accurate cognitive map development, compared to a non-AR control group. These findings suggest that AR may support cognitive training by reducing cognitive load and enhancing spatial encoding through multimodal cues. Although promising, these systems are still in their early stages of integration into cognitive aging research. Prior to the widespread adoption of these tools in clinical or home-based interventions, challenges related to hardware weight, field of view, environmental tracking, higher development costs, limited software availability, and difficulties integrating AR-based protocols into existing clinical workflows must be addressed. These factors have contributed to the slow rate of translation of AR into applied neuropsychological practice, despite its promise for ecological assessment in real-world contexts.

In contrast to non-immersive and semi-immersive VR, fully immersive HMD-based systems provide richer sensory engagement. Several studies have demonstrated the diagnostic sensitivity of HMD-based VR in detecting age-related spatial memory decline. The majority of the studies reviewed implemented iVR using HMDs such as the HTC Vive [30,34,47,54,67], Oculus Rift [38,39,44,62,65], or Meta Quest series [37,46,47,63,69]. These systems typically provide stereoscopic 3D environments, real-time motion tracking, and the potential for naturalistic body movement (e.g., walking or turning), thus providing a high degree of sensorimotor congruence and immersion. In particular, iVR platforms enable precise manipulation of spatial cues and allow for controlled simulation of real-world navigation scenarios, which is particularly valuable in assessing spatial memory. Several studies have demonstrated the capacity of iVR systems to detect subtle spatial impairments and facilitate cognitive training through embodied interaction (Figure 2). For example, Castegnaro et al. [33] found that an immersive object-location memory task differentiated between amnestic MCI and healthy aging more effectively than traditional neuropsychological assessments. In a related line of research, McCracken et al. [53] employed a homing task in iVR to simulate real-world navigation impairments, showing that older adults performed significantly worse when relying solely on visual or proprioceptive cues, but improved with multisensory integration. Meanwhile, Tuena et al. [64] implemented the ANTaging immersive platform for object-location training and observed enhanced long-term spatial recall in MCI participants following the intervention.

Ultimately, the choice of technological platform involves balancing ecological validity, participant tolerance, and research objectives. Higher immersion generally offers richer spatial engagement and diagnostic sensitivity. However, it must be noted that this increased engagement also necessitates greater technical resources and participant adaptability. 

### 3.4. Spatial Tasks and Cognitive Demands

Immersive technologies can precisely recreate real-world navigation challenges, allowing researchers to assess spatial cognition under conditions that closely mirror daily life while maintaining experimental control. A comprehensive review of the studies reveals the identification of four primary categories of spatial tasks, each engaging distinct cognitive processes and neural substrates. The most common tasks were wayfinding, involving route learning, landmark use, and decision-making. Path integration tasks focused on self-motion and egocentric orientation with minimal cues. Object-location tasks tested spatial binding linked to the hippocampus and entorhinal cortex. Finally, strategy tasks assessed the use of egocentric vs. allocentric navigation strategies, offering insights into cognitive flexibility and compensatory mechanisms (Figure 3).

The most common were wayfinding tasks, implemented in 18 studies (43%) [42,46,59,69,70]. These tasks entailed navigating toward a predetermined objective within virtual or real environments, frequently employing route learning, decision points, or landmark-based cues. For instance, Xu et al. [69] demonstrated that training in a virtual simulation of a real-world building significantly enhanced participants’ subsequent navigation performance in the corresponding physical environment, suggesting the potential of iVR for transfer of learning and ecological relevance.

A second major category included path integration paradigms, used in 13 studies (31%) [30,37,41,44,48,54]. These involved participants tracking their position and returning to a starting point after displacement, usually without the aid of external landmarks. This type of task is particularly demanding because it relies on self-motion cues, vestibular input, and internal spatial updating mechanisms. Newton et al. [54] provided significant evidence indicating that impairments in VR-based path integration are correlated with AD risk markers, including APOE ε4 genotype and CAIDE scores, highlighting its diagnostic potential.

In contrast, object-location memory was used in seven studies (17%) [33,55,57,63,64,65]. These required participants to encode and retrieve the positions of specific items in space. These tasks are closely linked to the functionality of the hippocampal and entorhinal cortex, both of which are affected early in AD. In an immersive VR intervention, Tuena et al. [64] reported significant gains in long-term spatial memory and mental rotation ability among MCI patients, suggesting that object-location memory tasks may both assess and support the rehabilitation of spatial deficits. 

Finally, four studies (9%) [35,38,58,60] explored the capacity of participants to engage and flexibly shift between egocentric (self-referenced) and allocentric (environment-referenced) frames of reference. This ability is known to decline with age and is particularly vulnerable in MCI. Castillo Escamilla et al. [35] observed that older adults with low working memory capacity had pronounced deficits in switching from egocentric to allocentric strategies, underscoring the role of executive function in spatial cognition and the utility of immersive tasks to reveal such deficits.

Collectively, these spatial task paradigms tap into distinct yet complementary cognitive processes, providing sensitive markers of early neurodegenerative changes and potential avenues for targeted cognitive training. 

### 3.5. Outcome Measures and Diagnostic Sensitivity

Studies employed a wide range of outcome measures, which can be grouped into four categories: behavioral outcomes, physiological outcomes, neuroimaging outcomes, and multimodal approaches. 

Behavioral outcomes included task completion times, navigation errors, distance or angular deviations, and success rates in locating objects or routes. These measures directly capture performance in spatial tasks and form the primary diagnostic metrics. Several immersive tools demonstrated superior diagnostic sensitivity compared to standard cognitive batteries. For instance, Bayahya et al. [31] demonstrated that VR scores (task completion time and correct responses) correlated with Mini-Cog and showed 97.2% accuracy in detecting dementia. Castegnaro et al. [34] showed that angular errors in iVR path integration tasks distinguished MCI+ from MCI– with high effect sizes. Da Costa et al. [38] and De Silva et al. [39] showed that SOIVET VR tasks could reliably differentiate MCI from controls. Additionally, these tasks exhibited strong correlations with visuospatial cognitive domains. Similarly, Chatterjee and Moussavi [36] found that spatial learning scores in a VR task predicted MoCA performance and cognitive status in AD and MCI patients. The high diagnostic utility of these immersive tools is complemented by their strong ecological validity. Unlike traditional paper-and-pencil tests, VR-based paradigms simulate real-world navigation challenges, making behavioral outcomes more representative of everyday cognitive functioning and real-world navigation. This alignment with real-life scenarios enhances their translational value and potential for clinical adoption.

Physiological outcomes provide complementary insights by detecting subtle cognitive load and stress responses. Heart rate variability and skin conductance responses have been used as indices of increased effort or disorientation. Amaefule et al. [29] demonstrated that spatial disorientation during VR navigation tasks was associated with increased gait variability and psychophysiological stress markers in older adults, suggesting elevated cognitive demand. Electroencephalogram (EEG)-based studies further highlight the role of neurophysiological markers. Hanert et al. [43] showed that patients with AD exhibited reduced sleep spindles and impaired slow oscillation–spindle coupling, which correlated with poor spatial memory consolidation.

Neuroimaging outcomes establish the link between behavior and brain structure or function. Functional Magnetic Resonance Imaging (fMRI) have been employed to complement performance metrics, revealing altered HC and RSC activity in older adults exhibiting reduced spatial learning abilities [40]. Castegnaro et al. [33] reported that object-location memory and path integration tasks in iVR were associated with entorhinal cortex and hippocampal volume, providing biological validation of task sensitivity.

Multimodal approaches combine behavioral, physiological, and imaging data to provide more comprehensive diagnostic profiles. The integration of these factors enables the detection of subtle impairments that might be missed by the use of behavioral tests alone and facilitates a differential diagnosis between normal aging and early neurodegeneration. Studies that employed multimodal outcomes, integrating behavioral, subjective, such as enjoyment, safety and comfort [37,62], and neurobiological data provided the most comprehensive profiles. Newton et al. [54] demonstrated that path integration error in VR predicted not only cognitive risk but also grid-cell activity patterns in the entorhinal cortex as measured by 7T fMRI.

Beyond the evaluation of diagnostic performance, several studies considered user experience factors, including enjoyment, safety, and comfort. These variables influence compliance, retention, and the validity of longitudinal assessments. For instance, De Silva et al. [39] reported low levels of discomfort and high tolerability among both healthy older adults and individuals with MCI using the SOIVET-Maze immersive system, with no significant group differences. Platforms designed to reduce cybersickness and enhance user comfort are more likely to be adopted in clinical settings, especially in older populations.

### 3.6. Interventions and Training Effects

While the exploration of immersive technologies has primarily focused on assessing spatial cognition, there has been a recent proliferation of studies that have begun to investigate their potential as rehabilitation tools of spatial memory in aging populations. Spatial memory, in particular, is an attractive target for intervention, given its central role in functional independence and its early decline in neurodegenerative disorders such as AD. VR offers a unique platform for repeated, adaptive, and ecologically valid training that may engage the neural systems implicated in spatial learning, navigation, and orientation.

Some studies included in this review implemented structured intervention protocols using immersive technologies to assess their spatial training effects [63,64,65,67,69]. These studies recruited digitally literate older adults, including MCI patients [64,65], with relatively high education levels, which limits generalizability. However, a notable exception exists. For example, Wen et al. [67] specifically targeted community-dwelling elderly with no prior VR or gaming experience, ensuring inclusion of individuals with lower digital literacy. 

The interventions were characterized by variability in duration, task design, and population characteristics. Collectively, these interventions investigated the feasibility of using virtual environments to improve specific aspects of spatial memory. For example, Tuena et al. [65] employed the ANTaging VR platform to train spatial memory in MCI patients. Participants engaged in object-location recall tasks within an immersive 3D environment that utilized egocentric and allocentric cues. Compared to those receiving traditional paper-and-pencil cognitive stimulation, the VR-trained group showed significant improvements in spatial mental rotation and long-term memory performance. This finding suggests that immersive, embodied interactions may enhance encoding and retrieval processes more effectively than static training methods or traditional cognitive stimulation. Similarly, Wen et al. [67] investigated the impact of virtual spatial training combined with EEG monitoring in healthy community-dwelling older adults. Their findings revealed significant post-training gains in spatial recall, supported by EEG-based neural correlates that distinguished pre- and post-intervention states. However, broader use of physiological, neuroimaging or biomarker data remains limited across intervention studies.

A particularly novel approach was taken by Sunami et al. [63], who integrated olfactory stimuli into an immersive VR platform. Older adults who engaged in olfactory-enhanced spatial memory tasks showed improvements in symbolic rotation and word-location recall. Notably, the study also highlighted the potential neuroplastic effects of olfactory VR, citing prior evidence of structural HC changes following olfactory training. The findings support the emerging potential of combining sensory modalities, such as visual, vestibular, and olfactory, to optimize cognitive interventions in VR. 

The majority of interventions were relatively brief in duration, often spanning only a few sessions. None of the reviewed studies incorporated long-term follow-up assessments, leaving the durability of cognitive benefits uncertain. Studies such as Sunami et al. [63] explicitly acknowledge this limitation, emphasizing the need for longitudinal research to evaluate sustained effects. Importantly, few studies examined whether training gains generalized to real-world navigation or daily functional tasks. While some improvements were observed in trained tasks, evidence of transfer to broader cognitive domains or real-life behaviors remains limited but promising. In the trial by Tuena et al. [64], immersive VR training showed significant improvements maintained at 3-month follow-up in both virtual performance and in standard neuropsychological tests, suggesting a potential for cognitive transfer to broader domains of spatial processing. Although the study did not directly assess functional navigation in daily life, the observed improvements in egocentric and allocentric performance could theoretically support better real-world orientation and independence. For instance, Xu et al. [69] designed a VR training protocol replicating a real university building and assessed its impact on actual wayfinding performance in that physical environment. The findings indicated that while VR training had limited impact on tasks that directly replicated the training content, it significantly improved performance in novel navigation tasks within the same real-world environment. This suggests that immersive VR may foster generalizable spatial learning and familiarity that translates to untrained but ecologically similar contexts. Nevertheless, it is imperative that subsequent research on intervention effects incorporate both subjective and objective assessments of daily navigation. 

Finally, several practical barriers were identified across the studies. These include cybersickness, especially in tasks involving active movement, equipment cost, substantial hardware requirements, calibration difficulties, and limited accessibility for users with mobility impairments. Such challenges have the potential to adversely affect participant engagement, particularly in home-based or longitudinal interventions.

### 3.7. Quality of Included Studies and Risk of Bias

In accordance with the JBI checklist (Appendix A), a total of 20 articles were classified as analytic cross-sectional studies, indicating an overall acceptable methodological quality [30,33,34,35,36,38,39,40,42,44,45,46,48,51,52,53,54,55,60,68], though with variability across domains. Most studies clearly defined inclusion criteria (Q1) and adequately described participants and settings (Q2). The measurement of exposures and outcomes (Q3, Q7) was generally reliable; however, in some cases, validity was not established. For instance, Koike et al. [48] and Kalantari et al. [46] did not validate the VR task employed. In contrast, Lowry et al. [51] employed the standardized Virtual Supermarket Task and examined its sensitivity and specificity for AD and vascular cognitive impairment, thereby reinforcing outcome validity (Q4, Q7). Confounding factors (Q5) and their management (Q6) were less consistently addressed. Kalantari et al. [46] explicitly noted the omission of potentially relevant covariates, such as gender. McCracken et al. [53] accounted for prior VR experience as a possible confounder, and Newton et al. [54] systematically adjusted for sex, age, and education, variables that are often underreported in demographic descriptions and seldom considered as modulators. The statistical analyses (Q8) were, in general, appropriate. However, the level of adjustment for confounders varied considerably across studies.

A substantial portion of the included studies fulfilled quasi-experimental criteria and demonstrated generally robust methodological quality, although several recurrent limitations were identified [37,49,50,56,61,62,63,64,65,67,70]. The majority of studies clearly defined causal relationships (Q1) and employed intra-subject or mixed designs that reduced interindividual variability and increased internal validity [37,49,70]. The utilization of mixed intra-/inter-subject approaches [50,61] has been demonstrated to enhance statistical power (Q2). Oliver et al. [56] further enhanced validity through short- and long-term assessments, while Tuena et al. [64,65] achieved the highest rigor by including external TAU controls, though attrition was not always fully addressed (Q8). In contrast, the study by Stramba-Badiale et al. [62] did not include baseline measures. Across studies, outcomes were consistently measured (Q6–Q7) and statistical analyses appropriate (Q9). In summary, most quasi-experimental studies achieved high internal validity through crossover and repeated-measures designs. However, external validity and reliance on non-traditional controls remain the main limitations.

Three randomized controlled trials were included, showing overall moderate-to-high methodological quality with heterogeneity across domains (Q2–Q13) [29,59,69]. Amaefule et al. [29] did not report allocation concealment or blinding but ensured high retention (96.2%) and reliable outcome measurement (Cohen’s kappa). Qiu et al. [59] similarly lacked details on randomization and blinding, with a small and homogeneous sample limiting external validity. However, the use of linear mixed-effects models and Cox regression provided robust statistical handling of repeated measures. In contrast, Xu et al. [69] employed a multi-arm double-blind design, intention-to-treat analysis, and advanced statistical adjustments (sample size planning with GPower, clustering effects), representing the highest methodological rigor.

Across the included diagnostic test accuracy studies [31,41,47,57,58], methodological quality can be considered moderate, with recurrent limitations related to unclear patient enrollment (Q1), inappropriate use of case–control design elements (Q2), and lack of transparency regarding blinding of index and reference test interpretation (Q4, Q7), all of which may compromise internal validity. Nonetheless, all studies employed appropriate reference standards (Q6), applied consistently across participants (Q9), and generally analyzed complete datasets (Q10), supporting reliability of outcome assessment. Among them, Park [57] provided the most rigorous design, uniquely ensuring blinded evaluation of all neuropsychological assessments, thereby minimizing review bias and strengthening the robustness of its findings relative to the others. It is important to note that several analytic cross-sectional studies, while not primarily designed for psychometric validation, also assessed the measurement properties of VR tasks as diagnostic instruments, providing convergent evidence of validity and reliability [33,34,36,38,39,42,51]. Specifically, ROC analyses demonstrated criterion-related validity [33,34,36,38], internal consistency and test–retest supported reliability [39], predictive validity was established against real-world navigation [42], and sensitivity/specificity were indirectly evaluated [51], collectively reinforcing the diagnostic potential of VR paradigms.

Only Hanyu et al. [21] conducted a cohort study showing a good methodological quality. Groups were comparable at baseline (Q1), exposures were reliably assessed with standardized MRI/SPECT (Q2–Q3) and validated cognitive and VR outcomes were applied consistently (Q6–Q7). Confounding was only partially addressed, with adjustment for age and education but not vascular comorbidities or MCI subtypes. The authors acknowledged this as a limitation (Q4–Q5). The follow-up was relatively brief, yet it was sufficient to detect early progression (Q8). Attrition was minimal (missing patients 2.7%; Q9–Q10), and statistical analyses were appropriate, though limited to conventional methods (Q11).

### 3.8. Limitations Across Studies

While the reviewed studies collectively demonstrate the potential of immersive technologies to enhance the assessment and training of spatial memory in aging populations, several methodological and practical limitations were consistently observed. The publication of positive or novel findings is more probable, resulting in a potential publication bias that may overestimate the effectiveness of immersive technologies for assessment and intervention. The scarcity of null or negative findings results in the literature that is disproportionately oriented towards optimistic outcomes. Limitations are also associated with the sample size and its representativeness, the technological barriers and usability challenges, the lack of longitudinal studies and biomarker validation, and the variability in the tasks designed, virtual environments, and interfaces. These limitations impact the interpretability, replicability, and scalability of findings. Addressing these limitations will serve two primary functions. Firstly, it will enhance the scientific rigor of future studies, thereby ensuring the credibility of the research and its findings. Secondly, it will accelerate the translation of virtual and mixed reality systems into scalable, real-world applications for public health and clinical care.

A major concern across the literature was the reliance on small, often homogenous samples. Many studies recruited individuals who were high-functioning, well-educated, or already familiar with technology [39,41,67]. This introduces selection bias and restricts the generalizability of findings to broader populations, particularly those with lower digital literacy or more advanced cognitive impairment. The underrepresentation of frail older adults, individuals from diverse socioeconomic backgrounds, and those with sensory or mobility impairments limits the external validity of intervention results and may obscure important usability challenges. Participant acceptability and accessibility remain under-explored, particularly in relation to individuals with reduced mobility or unfamiliarity with digital devices. The cultural adaptation of virtual environments, including consideration of architectural styles or navigation cues, has received minimal attention, despite its relevance for cross-cultural generalizability.

Despite the increasing accessibility of high-immersion virtual reality systems, significant usability concerns persist, particularly for older and clinical demographics. Navigation was implemented using a variety of input methods, such as joysticks, treadmills, teleportation, or real walking. However, the heterogeneity of the interface design may influence immersion levels, motor demands, and user engagement. These factors have the potential to influence cognitive performance and complicate the comparison of results across studies. Cybersickness, which is typically triggered by visual-vestibular mismatches or prolonged exposure, is characterized by dizziness, nausea, and disorientation. This condition was particularly prevalent during tasks involving active locomotion or visually rich environments [52,62]. Furthermore, hardware limitations, including constrained field of view, substantial equipment size, and calibration challenges, were identified as factors affecting user comfort and prolonged engagement. These issues become especially salient in the context of longitudinal or home-based interventions, where repeated exposure and minimal supervision are requisite.

While a few studies, including those by Newton et al. [54] and Castegnaro et al. [33], incorporated neuroimaging or genetic risk assessments, the majority lacked longitudinal designs or biological validation. Most interventions assessed outcomes immediately post-training, without implementing a follow-up strategy, making it difficult to determine the durability of cognitive gains or their predictive value for clinical progression. Moreover, the absence of biomarker-based classification in many MCI or AD samples raises concerns about diagnostic heterogeneity and the specificity of observed impairments. Few studies incorporated adequate blinding procedures or robust control groups, complicating the separation of true cognitive benefits from placebo effects or novelty-driven engagement. This is particularly salient in the case of older adults, in whom improvements may be indicative of increasing familiarity with VR hardware or interface, as opposed to genuine neurocognitive enhancement.

A further challenge is the considerable heterogeneity in spatial tasks, virtual environments, and user interfaces across studies. Navigation was implemented using a variety of inputs, ranging from joysticks, treadmills, teleportation, to real walking, limiting task comparability and user experience. The tasks included both abstract mazes [38,39,43,52] and photorealistic replications of real-world settings [45,47]. The outcome measures employed also varied, ranging from simple error rates to detailed behavioral and physiological metrics. This diversity complicates the establishment of comparisons across studies and limits the potential for meta-analytic synthesis. For instance, the cognitive demands of egocentric wayfinding in a two-dimensional environment may differ from allocentric route learning in a realistic cityscape, even if both are labeled as “navigation tasks”. The absence of standardized protocols and taxonomies further complicates the identification of the elements of task design that most significantly contribute to diagnostic sensitivity or training efficacy. 

Finally, the present review employed a qualitative synthesis approach. While some quantitative results were summarized (e.g., task performance, correlations with cognitive scores), no meta-analysis or formal effect size calculations were conducted due to methodological heterogeneity. A narrative approach was chosen to highlight trends, methodological gaps, and prevailing trends.

## 4. Discussion

This systematic review demonstrates the expanding role of immersive technologies in the assessment of spatial memory in aging populations. A comprehensive review of 42 studies published in the last five years revealed that diagnostic sensitivity, ecological validity, and user engagement vary significantly. These findings provide substantial evidence in support of the utilization of immersive environments as innovative tools for the detection of early spatial disorientation and cognitive decline, particularly in populations at risk for neurodegenerative conditions such as MCI and AD.

Regarding the advancing sensitivity and ecological validity in spatial assessment, one of the most significant observations from this review is the superior diagnostic sensitivity of iVR tasks compared to traditional neuropsychological tests. Studies such as those by Castegnaro et al. [33,34] and Newton et al. [54] demonstrated that iVR-based path integration and object-location memory tasks can differentiate between healthy older adults and those with MC. These tasks have also been shown to predict underlying neural markers, such as entorhinal cortex atrophy and AD risk scores. These results support the growing consensus that spatial memory impairments, particularly those linked to allocentric processing and navigation, can serve as early and specific indicators of cognitive decline. The ecological validity of immersive tasks was identified as a significant advantage. In contrast to conventional paper-and-pencil tests of spatial memory, iVR facilitates the simulation of complex, realistic environments that emulate navigational challenges encountered in everyday life. For instance, Xu et al. [69] reported that VR-based navigation training improved performance in real-world settings, suggesting that immersive tasks can facilitate the translation of laboratory assessment results into functional outcomes. Furthermore, tools like the SOIVET-Maze [38,39] and ANTaging [64] demonstrated not only diagnostic utility but also potential as rehabilitation platforms. However, the durability of these improvements remains unclear, as none of the studies included long-term follow-up assessments. Furthermore, while the ecological validity of immersive VR is a clear advantage, some evidence suggests that overly complex environments or excessive sensory input may lead to overstimulation or confusion in older adults with cognitive decline. As Buele et al. [71] highlight, such complexity has the potential to adversely impact task performance. In contrast, the utilization of simplified designs with intuitive interactions and clear cues may reduce cognitive load and improve usability in this population.

An emerging trend in the reviewed studies was the incorporation of multisensory [63] or embodied modalities [53,62] to enhance cognitive engagement and memory encoding. These studies corroborated the hypothesis that multimodal bodily interaction and proprioception can facilitate spatial learning and recall, thereby aligning with evidence on sensorimotor contributions to spatial memory. A review of studies that implemented iVR interventions, often with short-term training, resulted in improved spatial memory or navigation strategies. The findings from immersive interventions are consistent with the notion that spatial memory is plastic and trainable in older adults, including those with MCI. The inclusion of motivational elements, such as game-based goals or nostalgic content [56], also highlights the role of emotional engagement in facilitating adherence and efficacy. However, these studies did not report long-term follow-up.

Several limitations warrant consideration. A significant number of studies have been conducted with small sample sizes and non-representative cohorts, frequently comprising only high-functioning or highly educated older adults. This lack of diversity may bias results by overestimating the effectiveness of VR-based interventions, particularly among technologically literate users. Moreover, cybersickness and technological unfamiliarity persist as challenges to the widespread adoption of VR. Across the reviewed studies, the incidence of cybersickness was generally low [39,64,65], with no reported dropouts and minimal symptom scores, such as occasional dizziness and visual fatigue [63]. However, the occurrence of symptoms such as nausea, dizziness, and disorientation, often resulting from prolonged exposure or visual-vestibular mismatch, has been documented to lead to dropout rates ranging from 10 to 30% [27,33]. The heterogeneity in hardware, software platforms, spatial tasks, and outcome metrics complicates direct comparisons across studies. Despite the incorporation of neuroimaging or biomarker correlations in a select number of studies, the preponderance of research in this field is characterized by two notable deficiencies. Firstly, the majority of studies lack longitudinal follow-up, which is crucial for understanding the temporal progression of biological processes. Secondly, there is a dearth of neurobiological validation, a critical step in ensuring the reliability and validity of research findings. In particular, future studies should prioritize linking specific VR navigation metrics to structural or functional integrity of the hippocampus and entorhinal cortex, which are key regions involved in spatial memory and affected early in MCI and AD. 

This systematic review has several limitations that should be acknowledged. First, the literature search was restricted to three major databases (Web of Science, Scopus, and PsycINFO). Therefore, relevant studies indexed in other databases or unpublished data may have been missed, introducing potential publication and selection bias. Second, despite a comprehensive search strategy, variability in terminology across disciplines may have hindered the identification of all eligible studies. Such semantic heterogeneity is a common challenge in systematic reviews and could have affected the sensitivity of the search.

From a research perspective, it is recommended that future research prioritize longitudinal studies assessing the predictive value of VR-based spatial tasks for cognitive decline and functional outcomes. In addition, there is a critical need for standardized methodologies that facilitate cross-study comparisons. These methodologies should be inclusive of diverse participant samples, exhibiting a range of cognitive, physical, and technological profiles. A more comprehensive exploration of immersive rehabilitation frameworks is necessary, particularly those that integrate cognitive, emotional, and sensorimotor stimulation. Furthermore, the incorporation of multimodal biomarkers, including neuroimaging and physiological responses, into VR research holds potential to facilitate the early detection and personalized characterization of spatial memory decline. However, the integration of multimodal biomarkers into large-scale or home-based studies presents several practical challenges, including cost, accessibility, and participant burden. Additionally, there are ethical concerns regarding the utilization of sensitive clinical data outside of formal healthcare settings.

For clinical translation, the development of accessible, standardized VR protocols is essential for both clinical and home-based use. The incorporation of wearable and remote sensing technologies would facilitate real-time monitoring of spatial behavior in ecologically valid, everyday contexts. This approach supports the transition from laboratory-based cognitive testing to scalable, patient-centered solutions, thereby promoting the integration of immersive assessment tools into routine dementia care pathways. The development of standardized, accessible VR protocols is imperative for clinical and home settings.

## 5. Conclusions

In summary, immersive and semi-immersive VR technologies represent a transformative advancement in the assessment and rehabilitation of spatial memory among aging populations. The evidence synthesized in this review demonstrates their diagnostic sensitivity, ecological validity, and adaptability for both clinical and at-home use. Tasks involving path integration and object-location memory have demonstrated particular efficacy in the detection of early cognitive impairment and the support of personalized interventions. These tools have the potential to transform early diagnosis and intervention strategies for MCI and AD within real-world healthcare systems by enabling timely, ecologically valid, and engaging assessments.

While significant advancements have been made in the utilization of VR for assessment purposes, its growing role in cognitive training and rehabilitation also holds promise for enhancing spatial memory resilience and promoting functional independence. As challenges related to usability, standardization, and long-term validation are progressively addressed, these technologies are likely to become integral tools in preventive cognitive care. Nonetheless, significant limitations persist, including small sample sizes, limited longitudinal data, and the concern of cybersickness, which has the considerable potential to limit broader implementation.

The continued integration of multimodal biomarkers, wearable sensors, and user-centered design is imperative for the translation of research innovations into scalable public health solutions. To ensure a more extensive impact, future efforts must also prioritize the standardization of VR protocols and the inclusion of more heterogeneous sample of participants reflecting a range of cognitive, cultural, and technological backgrounds. These solutions must promote autonomy and cognitive resilience in older adults.

## Figures and Tables

**Figure 1 biomedicines-13-02105-f001:**
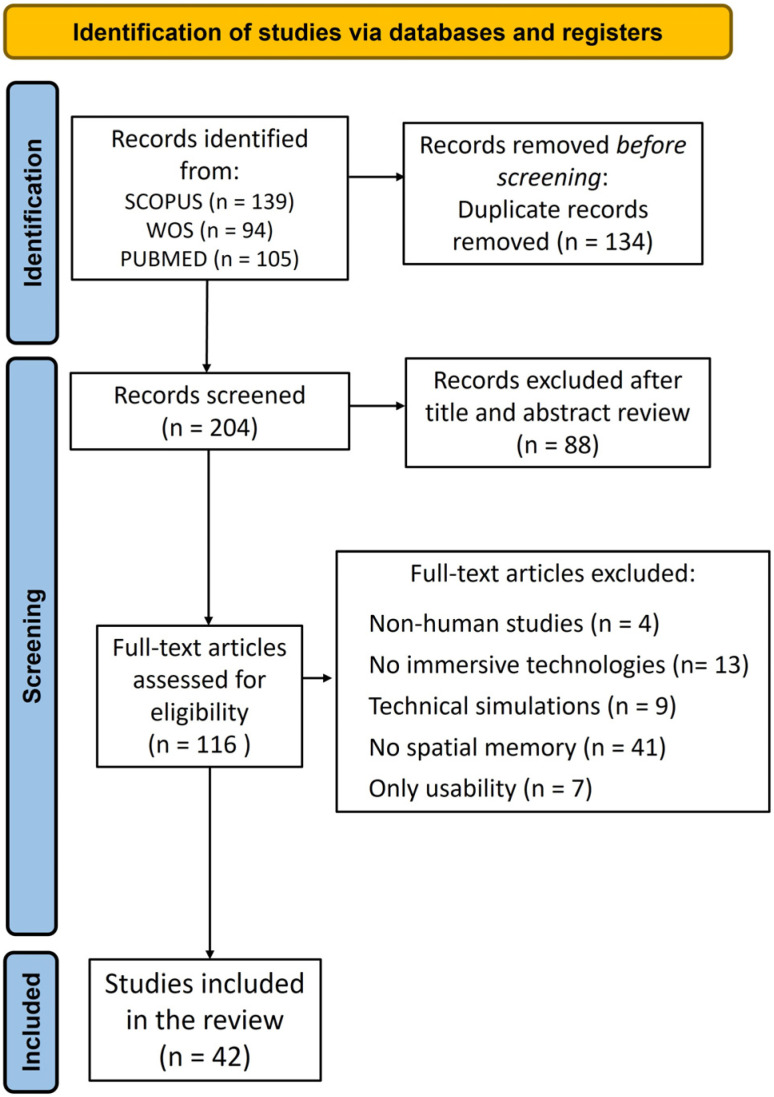
PRISMA flowchart of the search process.

**Figure 2 biomedicines-13-02105-f002:**
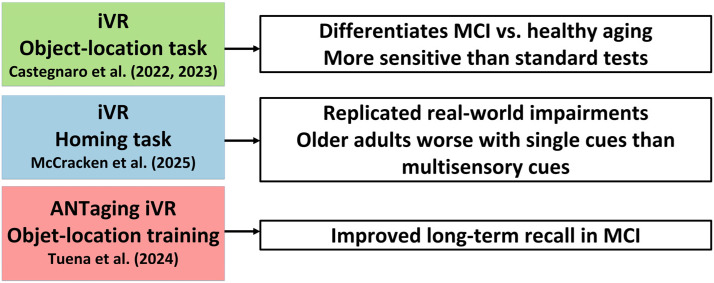
iVR Platforms: Enhanced Sensitivity for Detecting [33,34,53] and Training MCI [64].

**Figure 3 biomedicines-13-02105-f003:**
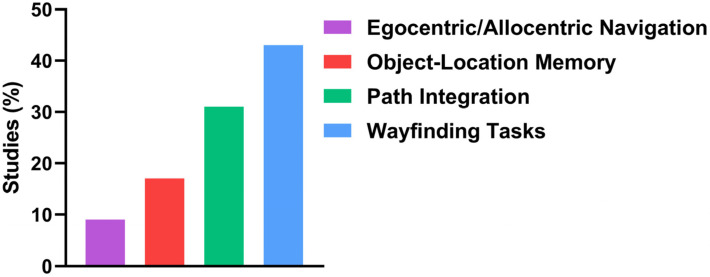
Types of spatial tasks implemented across the reviewed studies.

**Table 1 biomedicines-13-02105-t001:** Overview of the studies that employed immersive technologies for spatial memory assessment in aging populations.

Author(s), Year	Sample	Technology	Type of Task	Outcome Measures	Main Findings
Amaefule et al., 2023 [29]	N = 28 (14 control, 14 experimental); age: 60–85; MMSE * ≥ 29	Semi-immersive desktop-based VR * with a 180° projection screen and treadmill (GRAIL system); Immersion Level: Moderate	Wayfinding task with 14 decision points (7 crossings)	Behavioral: Gait parameters (speed, stride length, stance time, CVs *); Physiological: HRV *, SCR *	Disorientation increased gait variability and physiological responses, especially at crossings; induced disorientation was effective
Andac et al., 2024 [30]	N = 29 (14 glaucoma, 15 control); age: 42–80; MMSE ≥ 24	iVR with HTC Vive Pro; (walking in VR with motion-tracked controller); Immersion Level: High	Path integration (triangle completion with return pointing)	Behavioral: Travel time, Pointing time, Distance error; Ophthalmologic: MD *, pRNFL *	Glaucoma group showed longer travel and pointing times; performance correlated with MD and pRNFL; distance error was similar across groups
Bayahya et al., 2021 [31]	N = 115 (30 dementia, 20 MCI *, 65 controls); age: ≥ 50	Semi-immersive desktop-based VR using joystick control; Immersion Level: Low	Spatial navigation, orientation, visual memory, and delayed recall	Behavioral: Task completion time, accuracy (correct/incorrect responses); Statistical agreement with Mini-Cog; Visual memory, spatial orientation	VR scores correlated strongly with Mini-Cog (97.2% accuracy in detecting dementia)
Cammisuli et al., 2024 [32]	N = 40 (20 controls, 20 MCI due to AD *); age: 65–85	Active App Real-world Detour Navigation Test with Howdy Senior wearable sensor system; Immersion Level: Low	Egocentric and allocentric navigation in Real-world detour navigation (DNT-mv *)	Behavioral: Movements of hesitation, wrong turns, CBT *; Physiological: heart rate, respiration, motion via accelerometer	MCI patients showed deficits in egocentric and allocentric navigation that correlated with CBT and stress responses
Castegnaro et al., 2022 [33]	N = 100 (23 aMCI *; 24 controls (age: 60–80); 53 young (age: 20–30)); aMCI biomarker (9 positive, 7 negative)	iVR using HTC Vive; locomotion-based and handheld controller input Immersion Level: High	Object location memory, object recognition, object in context association	Behavioral: ADE *, % correct in recognition/context tasks; Neuroimaging: MRI * (alEC *, HC * volume)	aMCI patients had impaired spatial binding; object location memory task outperformed standard tests in distinguishing MCI; alEC volume predicted performance
Castegnaro et al., 2023 [34]	N = 110 (31 young (age: 20–35), 36 old (age: 60–85), 43 MCI (age: 60–85)); MCI subgroup: MCI+ * = 11, MCI− * = 14	iVR (HTC Vive); (free walking with real-world movement) Immersion Level: High	Path integration (triangle completion with return pointing)	Behavioral: distance and angular error	MCI+ group overestimated turning angle and showed increased angular noise; angular errors distinguish MCI+ from MCI− and controls
Castillo Escamilla et al., 2023 [35]	N = 60 (48 old (age: 60–80), 12 young (20–30))	Desktop-based VR; Joystick-controlled spatial navigation task (The Boxes Room); Immersion Level: Low	Egocentric and allocentric navigation in The Boxes Room	Behavioral: number of errors per condition and trial block, CBT, Digits, VPA *, MMSE; K-index	High-working-memory older adults performed similarly to young adults; low working memory older adults showed deficits in allocentric switching; performance correlated with working memory capacity
Chatterjee & Moussavi, 2025 [36]	N = 30 (10 young, 10 old, 10 AD); age: 20–88	Non-immersive desktop VR game (“Barn Ruins”); Joystick-controlled navigation; Immersion Level: Low	Wayfinding and path integration; route learning and recall in 3 difficulty levels	Behavioral: SLC (based on errors and difficulty), MoCA *, MADRS *, help button use	SLC strongly correlated with MoCA, predicted cognitive status, and distinguished between groups
Chen et al., 2024 [37]	N = 16 healthy old; age ≥ 60	iVR using Meta Quest Pro; Natural motion-based bike system (SilverCycling) vs. joystick control; Immersion Level: High	Path integration tasks: Intersection Direction Task and Landmark Sequence Task	Behavioral: Spatial orientation (task accuracy), SSQ *, IPQ *, NASA-TLX *, subjective ratings (enjoyment, safety, comfort)	SilverCycling improved spatial orientation in the Intersection Direction Task; users preferred it for enjoyment and safety
Da Costa et al., 2021 [38]	N = 48 (29 controls, 19 MCI); age: 61–92	iVR (Oculus Rift CV1); seated use with touch controllers; Immersion Level: High	SOIVET Maze (egocentric + survey to route), SOIVET Route (allocentric route learning)	Behavioral: Correct turns (Maze); Locations in correct order (Route); Correlation with ACE-R *, MRMT *, BJLO *, TOL *	Both tasks distinguished MCI from controls; Maze correlated with visuoperception, mental rotation and planning, Route with memory and visuoconstruction
De Silva et al., 2023 [39]	N = 43 (24 controls, 19 MCI); age: 61–92	iVR (Oculus Rift CV1); seated with touch controllers; Immersion Level: High	Egocentric orientation using SOIVET Maze (virtual navigation based on map guidance)	Behavioral: correct turns; Subjective: Cybersickness Questionnaire, Presence Questionnaire; Test–retest (ICC *)	High sense of presence and low cybersickness in both groups; good stability (especially in MCI); strong test–retest correlation; positive user experience
Diersch et al., 2021 [40]	N = 114 Behavior: (17 young (age: 21–28), 17 old (age: 61–72)); fMRI *: 25 young, 32 old (age: 58–75)	Desktop-based virtual environment (photorealistic VE * of historic city); Immersion Level: Low	Spatial learning and directional pointing (landmark-based navigation and cognitive map retrieval)	Behavioral: Absolute pointing error, reaction time; Neuroimaging: fMRI: BOLD * activation in HC and RSC *	Older adults showed reduced spatial learning, altered HC and RSC activity, and increased hippocampal excitability
Fu et al., 2022 [41]	N = 60 Healthy old (age: 24–84)	iVR using Oculus Quest; real walking in 3.5 m^2^ space; Immersion Level: High	Path integration (triangle completion with return to starting point without visual cues)	Behavioral: Absolute distance error, angle deviation, path deviation, return time, DST *; Neuroimaging: MRI: EC *, HC volumes	High reliability (performance declined with age; correlated with DST and EC thickness and HC volume)
Goodroe et al., 2025 [42]	N = 20 Healthy old (age: 54–74; MoCA ≥ 26)	Semi-immersive mobile app-based VR (Sea Hero Quest on tablets); Immersion Level: Low	Wayfinding tasks in virtual and real-world London streets	Behavioral: Distance travelled (virtual and real); self-reported Navigation Strategy Use; MoCA	Sea Hero Quest predicted real-world navigation at medium difficulty level; Navigation strategy correlated with virtual, not real-world performance
Hanert et al., 2024 [43]	N = 24 (12 early-stage AD, 12 controls); age: 53–85	Non-immersive desktop VR (Virtual Water Maze); joystick navigation; Immersion Level: Low	Allocentric spatial navigation (hidden treasure in Virtual Water Maze) Long-term retrieval	Behavioral: Relative dwell time in target quadrant; Physiological: EEG *-based SOs *, spindles, SO-spindle coupling	AD patients showed impaired verbal and reduced spatial memory consolidation, reduced fast spindle amplitude and fewer SO-spindle couplings during sleep
Hanyu et al., 2024 [44]	N = 71 MCI (14 Early MCI, 20 Late MCI– *, 37 Late MCI+ *); mean age: 74.4; n = 45 followed for 12 months	Immersive 3D VR with goggles and joystick; participants seated and rotated on swivel chair; Immersion Level: Moderate	Path integration (landmark-free navigation and return)	Behavioral: Distance error, angular error; MMSE, MoCA (baseline and 12-month change); Neuroimaging: AD brain score (MRI + SPECT *)	LMCI+ group showed greater path integration errors; path integration predicted 12-month cognitive decline
Hilton et al., 2020 [45]	N = 39 (20 young (mean age: 24), 19 old (mean age: 73))	Desktop-based photorealistic virtual environment (Virtual Tübingen); Eye-tracking + auditory probe; Immersion Level: Low	Route learning, direction recall, order memory	Behavioral: Route recall (direction test), order memory, auditory probe; Eye-tracking	Older adults showed reduced route learning; attentional measures predicted performance
Kalantari et al., 2024 [46]	N = 36 (18 young (age: 18–30), 18 old (>55 years))	iVR (Meta Quest 2 with joystick navigation) vs. identical real-world building; Immersion Level: High	Wayfinding across 7 tasks in a multilevel educational facility	Behavioral: distance traveled, task time, errors, backtracking, sign interaction, directional pointing; Subjective: workload, uncertainty, difficulty	VR led to longer paths, more errors, higher uncertainty, and greater workload than real-world; no age differences in VR vs. real outcomes
Kim et al., 2023 [47]	N = 46 (17 AD (age: AD 69 ± 8, MMSE: 21), 14 aMCI (age: 71 ± 7, MMSE: 25), 15 controls (mean age: 68 ± 8, MMSE: 29))	iVR (HTC Vive HMD * with hand controllers); photorealistic virtual living room; Immersion Level: High	HOT *: prospective, free recall, recognition, matching	Behavioral: HOT total and subtest scores; trajectory path, distance, duration, stay points; MMSE, SVLT *, RCFT *	HOT-differentiated AD, aMCI, and controls; scores aligned with standard tests; trajectories showed disorientation in AD and aMCI
Koike et al., 2024 [48]	N = 177 age: 20–89; subgroup analysis by decade	iVR (Meta Quest 2) (joystick + swivel chair navigation); Immersion Level: Moderate–High	Path integration (3-flag return task) and spatial cognition (hidden object search)	Behavioral: Error distance in path integration, quadrant dwell time in spatial memory; VR usability	Path integration declines from age 50 onward; greater error and variance in older adults; spatial memory preserved
Ladyka-Wojcik et al., 2021 [49]	N = 30 mean age = 75.8 ± 6; MoCA ≥ 20	Desktop-based virtual environments using Unity3D; egocentric (3D) and allocentric (2D map) views; Immersion Level: Low	Object location memory with egocentric and allocentric encoding and testing; frame switching task	Behavioral: Euclidean error in location recall; MoCA; self-reported Navigation Strategy use	Memory errors increased during frame switching (especially from egocentric to allocentric); higher MoCA and Navigation Strategy predicted better egocentric learning
Lokka & Çöltekin, 2019 [50]	N = 81 (42 young (age: 18–35), 39 old (age: 60–85)); MMSE ≥ 27	Desktop-based video walkthroughs of 3D virtual cities (Abstract, Realistic, and Mixed VE); Immersion Level: Low	Route learning with perspective switch (1st person to aerial); map recall	Behavioral: Route recall accuracy (maps); MRT *, VSM *; immediate and delayed recall (1 week)	Mixed VE improved spatial learning, especially for high MRT and VSM; older adults showed greater difficulty with perspective switches
Lowry et al., 2020 [51]	N = 39 (9 VCI *, 10 early AD, 20 controls (mean age: 77))	Tablet-based Virtual Supermarket Task (video-based VR); Immersion Level: Low	Egocentric and allocentric navigation; heading direction; spatial updating	Behavioral: Correct egocentric response, distance error (allocentric), heading direction; Clock test scores; ROC * curves	Egocentric impairments were specific to VCI and distinguished it from AD; allocentric deficits showed no group differences
McAvan et al., 2021 [52]	N = 27 (15 old (mean age: 74.3), 12 young (mean age: 20))	iVR (HTC Vive Pro, wireless HMD, foot tracking); Immersion Level: High (free ambulation)	Virtual Morris Water Maze navigation: learning, probe, cue manipulation	Behavioral: Distance error (target memory), strategy use (allocentric/beacon), motion metrics; Disorientation used before trials	Older adults showed reduced spatial precision but preserved strategy use (allocentric and beacon) and generalized to novel viewpoints
McCracken et al., 2025 [53]	N = 43 (24 young, 19 old; age: 19–73); MMSE > 24	iVR (Varjo VR-3 HMD) and matched real-world environment; walking; Immersion Level: High	Homing task (triangle completion using landmark and self-motion cues)	Behavioral: Homing accuracy (error in cm), variability, cue condition (vision, self-motion, both)	Age-related deficits replicated across both real and virtual tasks; older adults showed more errors with single-cue VR tasks, multisensory cues improved performance
Newton et al., 2024 [54]	N = 99 Middle-aged adults (age: 43–66); stratified by FH+ *, APOE ε4, CAIDE * risk	iVR (HTC Vive with wireless backpack); real walking in open-field triangle completion; Immersion Level: High	Path integration with 3 return conditions (baseline, no optic flow, no distal cues)	Behavioral: Location error, angular and distance error; Neuroimaging: 7T fMRI and structural MRI of EC, HC, and RSC	Errors predicted AD risk (FH+, APOE ε4+, CAIDE); no impairment in other cognitive domains; grid-like signal in EC correlated with accuracy
Noguera et al., 2020 [55]	N = 46 (26 salsa dancers, 20 non-dancers; age: 49–70)	Non-immersive desktop VR (Boxes Room spatial task); joystick navigation; Immersion Level: Low	Spatial memory (Boxes Room) navigate and recall hidden object locations	Behavioral: Number of errors (Boxes Room), ANT-I * latency and accuracy; verbal fluency, planning (Zoo test), K-BIT *	No group differences in spatial memory; dancers outperformed controls in executive function tasks
Oliver et al., 2024 [56]	N = 20 Mild to moderate AD (age: 63–83; MMSE ≥ 18)	Desktop-based VR (Unity3D); Joystick navigation; Personalized nostalgic vs. control landmarks; Immersion Level: Low–Moderate	Virtual route-learning with embedded nostalgic vs. control pictures; picture recognition and spatial memory tasks	Behavioral: Picture recognition and directional recall, positive/negative affect, self-esteem, self-continuity, social connectedness, meaning in life	Nostalgic landmarks did not affect spatial memory, but improved picture recognition, positive affect, self-esteem, self-continuity, and social connectedness
Park, 2022 [57]	N = 92 (36 MCI, 56 controls; age ≥ 65)	Desktop-based immersive VR using joystick navigation (Unity engine); Spatial cognitive training VR; Immersion Level: Low–Moderate	Path integration (navigate to and recall object locations)	Behavioral: Euclidean distance error in 10 trials; Test–retest reliability; MoCA * and BDT * from WAIS-IV *; ROC curves	Spatial training showed higher sensitivity and specificity than MoCA for detecting MCI; high test–retest reliability; strong concurrent validity
Puthusseryppady et al., 2022 [58]	N = 37 (16 AD patients; 21 age/gender-matched controls; age: 50–80); community-dwelling	Non-immersive VR on iPad: VST * and SHQ *; Immersion Level: Low–Moderate	Egocentric and allocentric navigation (VST, SHQ); Detour navigation in real-world neighborhoods (DNT *)	Behavioral: egocentric and allocentric accuracy, route errors, disorientation moments, wayfinding distance/duration; DNT score	AD patients showed navigation impairments in both VR and real-world; SHQ wayfinding predicted DNT disorientation, VR lacked reliability for high-risk classification
Qiu et al., 2024 [59]	N = 32 (Conditions: 16 AR * and 16 controls); age: 60–75	AR using HoloLens 2 smartglasses (visual + auditory overlays); Immersion Level: Moderate	Indoor landmark-based navigation tasks (10 wayfinding trials)	Behavioral: Task completion time, distance traveled, pointing error, distance estimation error, sketch map scores; SUS *, MEC-SPQ *, SART *, NASA TLX	AR showed better wayfinding speed, shorter distance, and superior cognitive map development; benefits persisted after AR use ended
Rinne et al., 2022 [60]	N = 77 (26 children (age: 7–16); 32 young (age: 18–35); 19 old (age: 63–81)); balanced gender	Desktop-based VR; Unreal Engine; mouse + keyboard navigation; Immersion Level: Low	Egocentric (local cues) vs. allocentric (global cues) navigation tasks in a virtual landscape	Behavioral: Number of wayfinding errors per trial (1–6), learning rate, age-related performance	Wayfinding improved with age but declined in older adults, with greater allocentric than egocentric decline
Shayman et al., 2024 [61]	N = 44 (24 young (age: 19–30); 20 old (age: 61–78))	iVR (Varjo VR-3 HMD); free walking homing task with 3 cue conditions + 1 conflict; Immersion Level: High	Homing (triangle completion); return to start using vision, self-motion, or both	Behavioral: Homing accuracy, variability, cue weighting (observed vs. predicted); model fit	Older adults were less accurate and consistent in unisensory tasks; both groups improved with multisensory cues; older adults integrated cues suboptimally
Stramba-Badiale et al., 2024 [62]	N = 7 MCI (mean age: 75)	ANTaging software: iVR (Oculus Rift S) vs. Semi-immersive (Samsung UHD 4K monitor) with 3dRudder; Immersion Level: Moderate–High	Spatial memory: encoding and recall (object-location task); allocentric vs. egocentric cues	Behavioral: Usability (SUS, ITC-SOPI *), spatial error (distance and angle), task time, motor data (3dRudder); qualitative interviews	No error differences between systems; semi-immersive was more usable with fewer side effects; both aided memory training
Sunami et al., 2025 [63]	N = 30 Older adults; age: 63–90	iVR (Meta Quest 3) with olfactory display (12 scents via solenoid valves); Immersion Level: High	Olfactory-enhanced spatial tasks (object-location, odor recall, symbolic rotation)	Behavioral: HRT * (symbolic rotation), Object/Word Spatial Memory, Odor Identification, Missing Number Task, MMSE	HRT and Word Spatial Memory improved with short intervention; no changes in MMSE, visual memory, or Odor Identification
Tuena et al., 2024 [64]	N = 30 MCI patients (16 Usual-treatment; 14 ANTaging; mean age: 75); mostly aMCI	iVR (CAVE system + 3dRudder foot controller + 3D glasses); Immersion Level: High	Object-location recall using egocentric and allocentric cues (ANTaging task); compared to usual-treatment paper-based tasks	Behavioral: CSS *, MT *, story recall, MMSE, virtual spatial memory (Euclidean error)	ANTaging group showed significant improvement in spatial mental rotation (MT) and long-term spatial memory (CSS); virtual performance improved across sessions
Tuena et al., 2024 [65]	N = 15 MCI patients (10 aMCI; 5 naMCI *; mean age: 75.58 ± 5.3)	Five virtual interfaces: immersive VR (Oculus Rift S + 3dRudder), desktop VR; Immersion Levels: Low–High	Object-location recall in landmark-based virtual arena with egocentric and allocentric cues	Behavioral: Spatial memory error (Euclidean distance); MMSE, CSS, CBT, FAB *, TMT *, FCSRT *, RCPM *, GDS *	Bodily (immersive) and interactive allocentric map conditions improved spatial memory more than passive or cue-free navigation; free navigation impaired allocentric memory
Wang et al., 2025 [66]	N = 146 (44 young (age: 18–34); 53 young-old (age: 60–74); 49 old-old (age: 75–89))	Desktop-based interactive 3D VR; keyboard-controlled; Immersion Level: Low–Moderate	One-trial DMTP * task	Behavioral: Latency, pathlength, % misses, stationary time (% freeze), convex hull area, probe test proximity curves, MoCA	Aging impaired spatial working memory, especially in probe-based memory expression (weaker V-shaped preference for goal); MoCA predicted probe performance better than age
Wen et al., 2023 [67]	N = 7 healthy community-dwelling older adults (mean age: 67 ± 6.81)	iVR (HTC Vive Focus); Immersion Level: High	Virtual community training + city roaming test; spatial learning and recall via VR tasks	Behavioral: Recall and retrace the original route, GZSOT *, PTSOT *, CBT; CNN * model classification (pre vs. post); Physiological: EEG: PCMICSP *-based spatial features	Virtual training was effective in stimulating spatial learning and recall and spatial scales (CBT, GZSOT, PTSOT); EEG distinguished pre and post training
Wiener et al., 2020 [68]	N = 81 (37 young (age: 18–32); 44 old (age: 60–82)); ACE-III * > 82	Desktop-based VR (Unity3D); Passive navigation using keyboard; Immersion Level: Low	Route-repetition, Route-retracing, Directional-approach tasks	Behavioral: Correct responses; Response times	Younger adults outperformed older; only young group showed learning in route-retracing; performance declined with greater misalignment in directional-approach task
Xu et al., 2025 [69]	N = 49 (17 VR; 16 Video; 16 Control; mean age: 71)	iVR (Meta Quest 2); Unreal Engine 3D model of real building; seated teleportation; Immersion Level: High	Wayfinding in a virtual training environment with real-world post-training navigation tasks	Behavioral: Task duration, Distance traveled, Pointing error, Distance estimation, Spatial anxiety, Workload	VR training improved wayfinding in real-world tasks, but not in trained ones; no group differences in pointing/distance error
Zuo & Zhou, 2024 [70]	N = 48 (24 young (age: 19–27), 24 old (age: 60–83))	Desktop VR with mouse + button box navigation; Immersion Level: Low–Moderate	Outdoor–indoor wayfinding in 4 virtual scenarios; cognitive map drawing	Behavioral: Wayfinding time, hesitation count, turn errors; navigation interaction; regression frequency/angle; cognitive map drawing accuracy/time	Older adults showed more navigation interaction but lower map accuracy

* Abbreviations (in alphabetical order): ACE-III = Addenbrooke’s Cognitive Examination; ACE-R = Addenbrooke’s Cognitive Examination, revised version; AD = Alzheimer’s Disease; ADE = Absolute Distance Error; alEC = anterolateral Entorhinal Cortex; aMCI = amnestic Mild Cognitive Impairment; ANT-I = Attentional Networks Task I; AR = Augmented Reality; BDT = Block Design Test; BJLO = Benton Judgment of Line Orientation; BOLD = Blood Oxygen Level Dependent; CAIDE = Cardiovascular Risk Factors, Aging, and Incidence of Dementia; CBT = Corsi Block Test; CNN = Convolution Neural Network; CSS = Corsi Supra-Span; CVs = Coefficients of Variation; DMTP = Delayed Matching-To-Place; DNT = Detour Navigation Test; DNT-mv = Detour Navigation Test, modified (during naturalistic spatial navigation); DST = Digit Span Task; EC = Entorhinal Cortex; EEG = Electroencephalogram; FAB = Frontal Assessment Battery; FCSRT = Free and Cued Selective Reminding Test; FH+ = Familial Hypercholesterolemia; fMRI = functional Magnetic Resonance Imaging; GDS = Geriatric Depression Scale; GZSOT = Guilford–Zimmerman Spatial Orientation Test; HC = Hippocampus; HMD = Head-Mounted Display; HOT = Hidden Object Test; HRT = Hiragana Rotation Task; HRV = Heart Rate Variability; ICC = Intraclass Correlation Coefficient; IPQ = Igroup Presence Questionnaire; ITC-SOPI = ITC-Sense of Presence Inventory; K-BIT = Kaufman Brief Intelligence Test; MADRS = Montgomery–Asberg Depression Rating Scale; MCI = Mild Cognitive Impairment; MCI+ = Mild Cognitive Impairment with neuropathology; MCI− = Mild Cognitive Impairment with neuropathology; MD = Mean Deviation measure of visual field loss; MEC-SPQ = MEC Spatial Presence Questionnaire; MMSE = Mini Mental State Examination; MoCA = Montreal Cognitive Assessment; MRI = Magnetic Resonance Imaging; MRMT = Money Road Map Test; MRT = Mental Rotation Test; MT = Manikin Test; naMCI = non-amnestic Mild Cognitive Impairment; NASA-TLX = NASA Task Load Index; PCMICSP = Permutation Conditional Mutual Information Common Space Pattern; pRNFL = peripapillary Retinal Nerve Fiber Layer; PTSOT = Perspective Taking Spatial Orientation Test; RCFT = Rey Complex Figure Test; RCPM = Raven’s Colored Progressive Matrices; ROC = Receiver Operating Characteristic; RSC = Retrosplenial Cortex; SART = Sustained Attention to Response Test; SCR = Skin Conductance Response; SHQ = Sea Hero Quest; SLC = Spatial Learning Score; SOs = Slow Oscillations; SPECT = Single Photon Emission Computed Tomography; SSQ = Simulator Sickness Questionnaire; SUS = System Usability Scale; SVLT = Seoul Verbal Learning Test; TMT = Trail Making Test; TOL = Tower of London test; VCI = Vascular Cognitive Impairment; VE = Virtual Environment; VPA = Verbal Paired Associates; VR = Virtual Reality; VSM = Visuospatial Memory test; VST = Virtual Supermarket Test; WAIS-IV = Wechsler Adult Intelligence Scale—Fourth Edition.

## Data Availability

Not applicable.

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
