# Peer review of "Immersive Technologies Targeting Spatial Memory Decline: A Systematic Review"

_biomedicines, 2025, doi:10.3390/biomedicines13092105_

Round 1
Reviewer 1 Report
Comments and Suggestions for Authors
The authors summarized the most recent studies on the applications of emerging techniques (VR, AR) in detecting cognitive impairments (MCI and AD). Systematic study selection method, under the guiding of PRISMA, gives us an unbiased whole view of the frontier in this field. The main conclusion is that innovative technologies like VR show promising advantages in the diagnostic sensitivity and ecological validity for the assessment of spatial memory in aging groups. With a detailed investigation of 42 studies from multiple aspects, including sample characteristics, immersive level, spatial memory task, outcome measures and diagnostic sensitivity, the authors also pointed out the current limitations and provided constructive suggestions for the future study. Overall, this is a right-on-time review paper in an emerging field and I believe it will attract attentions and inspire more studies.
Here I have one minor suggestion on the illustration of the selected studies. It would be very helpful if the authors could be able to remake the published figures to demonstrate the conclusions. For example, in section 3.3, the authors used results from Castegnaro et al., McCraken et al., and Tuena et al. to demonstrate that iVR systems show better sensitivity in detecting MCI. It would be more straight-forward for readers to get the point if they can summarize the related results and reorganize them into a single figure.
Author Response
Authors: We thank the reviewers for their insightful and constructive comments, which have greatly helped us to improve the manuscript. Below, we provide a detailed point-by-point response. All revisions are marked in the revised version of the manuscript highlighted in yellow. We have implemented all the proposed modifications.
Reviewer 1:
The authors summarized the most recent studies on the applications of emerging techniques (VR, AR) in detecting cognitive impairments (MCI and AD). Systematic study selection method, under the guiding of PRISMA, gives us an unbiased whole view of the frontier in this field. The main conclusion is that innovative technologies like VR show promising advantages in the diagnostic sensitivity and ecological validity for the assessment of spatial memory in aging groups. With a detailed investigation of 42 studies from multiple aspects, including sample characteristics, immersive level, spatial memory task, outcome measures and diagnostic sensitivity, the authors also pointed out the current limitations and provided constructive suggestions for the future study. Overall, this is a right-on-time review paper in an emerging field and I believe it will attract attentions and inspire more studies.
Here I have one minor suggestion on the illustration of the selected studies. It would be very helpful if the authors could be able to remake the published figures to demonstrate the conclusions. For example, in section 3.3, the authors used results from Castegnaro et al., McCraken et al., and Tuena et al. to demonstrate that iVR systems show better sensitivity in detecting MCI. It would be more straight-forward for readers to get the point if they can summarize the related results and reorganize them into a single figure.
Response: We have created a new summary figure (new Figure 2) in the revised manuscript) that visually integrates the findings of Castegnaro et al., McCracken et al., and Tuena et al., highlighting how immersive VR systems show higher sensitivity in detecting MCI.
Reviewer 2 Report
Comments and Suggestions for Authors
The manuscript entitled “Immersive Technologies Targeting Spatial Memory Decline: A Systematic Review' addresses a highly relevant and timely topic at the intersection of neuropsychology, gerontology and immersive technologies. Its systematic review of VR and MR applications for assessing and rehabilitating spatial memory in ageing populations provides valuable insights for research and clinical practice alike.
I have made some suggestions on how you could improve your work. You don't have to agree with them or rewrite your work in the same way. They are simply intended to help you see things from a different perspective.
1) In the 'Introduction' section, the definition of 'spatial memory' is given twice: once on lines 42-44, and again on lines 49-53. This could be condensed to avoid repetition. Some parts explaining ecological validity and real-world navigation tasks are repetitive and could be shortened to maintain the text's flow.
2) Providing brief epidemiological context on the prevalence of MCI and AD in ageing populations would emphasise the urgency of developing new diagnostic tools.
3) Mentioning how current guidelines or clinical practice incorporate (or fail to incorporate) spatial memory testing would position the review within a practical healthcare framework.
4) Adding a note on how immersive technologies compare with other emerging digital tools (e.g. tablet-based cognitive tests and AI-driven diagnostics) would provide a fuller picture of the technological landscape.
5) 'Immersive technologies' are introduced early on (line 77) – it may be worth defining VR, iVR and MR briefly in one concise sentence so that non-specialist readers immediately understand the distinctions.
6) The terms 'diagnostic sensitivity' and 'intervention efficacy' could be introduced with a brief example to make the text more accessible to a broader scientific audience.
7) Please, consider separating 'assessment' and 'rehabilitation/training' into two distinct subsections within the introduction to improve clarity.
8) The 'Materials and Methods' section mentions the period from January 2020 to March 2025. Explaining why this range was chosen would strengthen the rationale. While limiting the search to English is common, this should be justified (e.g. accessibility or resources for translation).
9) Please, indicate whether studies including mixed-age populations were included, and whether data for participants aged 50 years or over could be extracted separately.
10) The screening process states that two reviewers screened studies independently, with a third reviewer resolving conflicts. It would be useful to state whether a standardised data extraction form or protocol was used to ensure consistency.
11) Figure 1 is referenced, but the text could briefly describe the numbers at each stage (e.g. duplicates removed, abstracts screened, full texts assessed, reasons for exclusion).
12) The section does not mention how the data were extracted (e.g. participant characteristics, interventions, outcomes). It does not describe the risk of bias or quality assessment of the included studies. Including this information would strengthen the methodological rigour of the review (e.g. by using the Cochrane Risk of Bias Tool, the Joanna Briggs Institute checklist, or a similar tool).
13) In Section 3.1 ('Included studies'), please indicate the balance between cross-sectional, longitudinal and intervention-based studies. Since 'variety of geographic locations' is mentioned later, it would be helpful to specify here how global the sample is (e.g. the number of continents or countries represented). Mention briefly whether most studies used VR, AR or mixed modalities to frame the rest of the review. The section would be more complete with a short statement summarising the main inclusion and exclusion criteria for selecting these 42 studies.
14) In Section 3.2 'Sample characteristics of included studies', you could add the following: a) Mean age and standard deviation across studies; b) Proportion of participants with higher vs. lower education levels; c) Representation of ethnic or cultural backgrounds (if available). You could also quantify roughly how many studies or participants fell into each cognitive category (e.g. % MCI, % AD, % healthy controls). While the lack of reporting is noted, it would be useful to place a stronger emphasis on how this gap impacts generalisability. If data exist, indicate whether participant attrition or recruitment bias was common (e.g. recruitment from clinics vs community samples). Mention whether control and clinical groups were typically balanced in number or if disparities were common.
15) The first sentence of Section 3.3 lists low-immersion desktop VR, high-immersion HMD VR, semi-immersive desktop VR and AR in quick succession. This could be reorganised into a clearer paragraph, starting with the least immersive and moving to the most immersive. Briefly define desktop VR, semi-immersive VR, fully immersive VR (HMD) and AR right at the start, so that the reader has a mental framework before reading the details.
16) Multiple sentences in the HMD section reiterate that these systems offer stereoscopic 3D and motion tracking, allowing for realistic body movement. This information could be condensed into a single clear sentence before moving on to examples.
17) Currently, examples from Castegnaro, McCracken and Tuena are given in separate sentences. These could be grouped together under the heading 'Several studies have demonstrated...' and their key findings could then be listed in a compact comparative way.
18) While the HMD section is very detailed, the desktop/semi-immersive section could be slightly expanded with an extra example of a practical application outside the lab to balance the depth of coverage.
19) The AR section is short and might benefit from some discussion of why it has not been more widely adopted (beyond hardware weight and field of view), for example cost, limitations in software development, or integration with clinical workflows.
20) Using transition phrases such as 'In contrast' or 'By comparison' between immersive HMDs, semi-immersive systems and AR would make the reader's journey smoother. Currently, the transition from HMDs to desktop VR feels abrupt.
21) After describing all platforms, conclude with a synthesis sentence that positions them along the immersion spectrum and links immersion level to research applicability. For example: 'Ultimately, the choice of technological platform involves balancing ecological validity, participant tolerance and research objectives. Higher immersion generally offers richer spatial engagement, but requires greater resources and technical support.'
22) The first two sentences of Section 3.4 are strong, but somewhat dense. You could make them more engaging by starting with a direct statement about why immersive technologies are uniquely suited to assessing spatial cognition (e.g. 'Immersive technologies can precisely recreate real-world navigation challenges, allowing researchers to...').
23) The list of the four main task categories is good, but it would be even better if you briefly linked each one to its main cognitive demand in the same sentence, to give the reader a 'map' before the detailed descriptions.
24) Currently, each task type starts abruptly ('Wayfinding tasks...', 'Thirteen studies...'). Consider adding linking phrases such as 'The most common were...', 'A second major category...', 'In contrast...', 'Finally...'. This would help to keep the reader oriented in the progression from the most to the least frequently studied task type.
25) In the object-location memory section, the phrase 'hippocampal and entorhinal cortex functioning' could be simplified to 'hippocampus and entorhinal cortex' to avoid repetition of 'functioning'. Similarly, 'may not only assess but also support' could be tightened to 'may both assess and support'.
26) Currently, the 3.4 section concludes with the egocentric vs. allocentric strategy example, which is interesting, but does not provide a summary of the four-task overview. Adding a final synthesis sentence could link the categories back to their diagnostic and rehabilitative potential, for example: 'Collectively, these spatial task paradigms tap into distinct yet complementary cognitive processes, providing sensitive markers of early neurodegenerative changes and potential avenues for targeted cognitive training.'
27) The '3.5. Outcome Measures and Diagnostic Sensitivity' lists many measures, but does not explain why each category (behavioural, physiological and neuroimaging) is important in the context of VR cognitive diagnostics. It is suggested that a brief note is added to explain how these measures contribute to early detection or differential diagnosis (e.g. physiological measures detect subtle stress responses and neuroimaging links behaviour to neural substrates).
28) Behavioural measures are clearly explained, but physiological and neuroimaging measures are only briefly mentioned. Please suggest adding at least one more concrete example of a physiological or EEG-based study to balance coverage.
29) The phrase 'several immersive tools demonstrated superior diagnostic sensitivity' could be expanded upon with a brief comparison — how much better were they compared to standard tests, if the data is available? If possible, please note the sensitivity or specificity percentages or effect sizes from the cited studies.
30) Enjoyment, safety and comfort are mentioned, but their role in diagnostic interpretation is not explained. Please, clarify whether these factors influence test performance, compliance or validity.
31) The section would be clearer if it were grouped into the following categories: (a) behavioural outcomes, (b) physiological outcomes, (c) neuroimaging outcomes and (d) multimodal approaches.
32) You could also consider mentioning ecological validity, i.e. how closely VR measures reflect real-world navigation and cognition, as this is an important reason for adopting these methods over traditional ones.
33) The '3.6. Interventions and Training Effects' section states that most interventions were brief, but does not address whether any studies evaluated the longevity of training effects following the intervention period (i.e. follow-up testing). Including such data would clarify the potential for long-lasting cognitive benefits.
34) It would also be valuable to discuss whether improvements in spatial memory achieved through VR transfer to functional tasks in real-world navigation, daily living or independence, especially given that this is a key goal for ageing populations.
35) While EEG findings are mentioned, other neuroimaging or physiological markers (e.g. fMRI connectivity changes or neuroplasticity biomarkers) from intervention studies could be highlighted to demonstrate why VR might be effective.
36) It would also be useful to briefly note whether any intervention studies included diverse participant groups (e.g. those with varying cultural backgrounds, levels of digital literacy or comorbidities), as this affects generalisability.
37) Including a sentence on practical barriers (e.g. cybersickness, equipment cost and accessibility for mobility-impaired older adults) would make the review more balanced and realistic.
38) Some sentences, particularly those introducing VR's theoretical potential (lines 341–345), could be condensed slightly to avoid repetition with earlier sections describing VR's ecological validity.
39) The phrase 'may enhance encoding and retrieval processes' (lines 356–357) appears more than once in different forms; one instance could be reworded or omitted for concision.
40) In the '3.7. Limitations across studies' section, please note that studies with positive or novel findings are more likely to be published, which could skew the literature towards optimistic outcomes. The scarcity of null or negative findings may overestimate the effectiveness of VR.
41) Some studies may lack robust control groups or adequate blinding, which makes it difficult to distinguish VR-specific benefits from placebo, novelty or engagement effects. In older adults in particular, performance improvements may partly reflect familiarity with the hardware or software rather than genuine spatial memory enhancement. Few studies explicitly separate these effects.
42) While VR offers ecological realism, some highly realistic environments may introduce uncontrolled variability, which complicates the interpretation of cognitive outcomes. Cultural familiarity with environmental cues (e.g. building styles or street layouts) could influence navigation performance yet this is rarely considered in task design.
43) Some details about input device types (e.g. joysticks, treadmills, teleportation and walking) could be condensed into a single sentence, as these specifics might be more suited to the methods review than the limitations section. The focus here should be on why the heterogeneity matters, rather than providing an exhaustive list of devices.
44)The lines about cybersickness prevalence (397-399) could be tightened slightly without losing clarity by summarising the symptoms and triggers more concisely, for example.
45) There is no mention of how the data were synthesised, or whether this was qualitative, quantitative, or both. If a meta-analysis or effect size calculation was considered, this should be mentioned briefly. Even if only a narrative synthesis was performed, stating the approach would add clarity.
46) The discussion of limitations and challenges (lines 102–106) could be expanded to include participant acceptability, accessibility for people with mobility impairments and the cultural adaptation of VR environments.
47) In the 'Discussion' section, 'MC' should likely be 'MCI' (mild cognitive impairment) in lines 437-438 for consistency.
48) The review notes 'improved performance in real-world settings' after VR training, but does not clarify whether these improvements were sustained. This could be explicitly tied to the limitation regarding the lack of long-term follow-up.
49) Although the discussion mentions small sample sizes and lack of diversity, it could highlight how these factors might bias the results (e.g. overestimation of VR efficacy in technologically literate participants).
50) The statement on cybersickness could be improved with the inclusion of data, such as incidence rates or its effect on dropout rates.
51) While neurobiological validation is identified as scarce, the section could specify which types of validation are most urgently needed (e.g. establishing a link between specific VR navigation metrics and the integrity of the hippocampus and entorhinal cortex).
52) The review mentions multimodal biomarkers (fMRI, PET, etc.) in its recommendations but does not discuss the practical or ethical challenges of integrating these in large-scale or at-home settings. Adding this information would strike a balance between feasibility and aspiration.
53) While the ecological validity of VR is praised, the section could acknowledge that excessive realism or complexity could overwhelm older adults with cognitive decline, potentially impacting their test performance.
54) The final paragraph combines recommendations for research (e.g. longitudinal studies and diverse samples) with recommendations for clinical translation (e.g. standardised protocols and wearable sensors). Separating these into two distinct subthemes would improve clarity.
55) In the 'Conclusions' section, briefly reiterate how VR assessments could transform early diagnosis and intervention strategies for MCI and AD within real-world healthcare systems.
56) While the 'Discussion' section acknowledges the role of VR in training and rehabilitation as well as assessment, this is absent from the conclusions.
57) Including a brief reminder that small sample sizes, a lack of longitudinal data and cybersickness are still barriers would make the conclusion more balanced.
58) Including one sentence calling for the standardisation of VR protocols and the inclusion of more diverse participant groups would mirror the strong recommendations in the discussion.
This manuscript contains valuable findings, but improving its clarity, the depth of its discussion and its methodological detail would further enhance its impact. I recommend accepting the manuscript subject to some revisions.
Comments on the Quality of English LanguageThe English could be improved to more clearly express the research.
Author Response
Authors: We thank the reviewers for their insightful and constructive comments, which have greatly helped us to improve the manuscript. Below, we provide a detailed point-by-point response. All revisions are marked in the revised version of the manuscript highlighted in yellow. We have implemented all the proposed modifications.
Reviewer 2:
The manuscript entitled “Immersive Technologies Targeting Spatial Memory Decline: A Systematic Review' addresses a highly relevant and timely topic at the intersection of neuropsychology, gerontology and immersive technologies. Its systematic review of VR and MR applications for assessing and rehabilitating spatial memory in ageing populations provides valuable insights for research and clinical practice alike.
I have made some suggestions on how you could improve your work. You don't have to agree with them or rewrite your work in the same way. They are simply intended to help you see things from a different perspective.
1) In the 'Introduction' section, the definition of 'spatial memory' is given twice: once on lines 42-44, and again on lines 49-53. This could be condensed to avoid repetition. Some parts explaining ecological validity and real-world navigation tasks are repetitive and could be shortened to maintain the text's flow.
Response 1: The introduction has been modified to provide a concise explanation of the two definitions of spatial memory, while eliminating redundant descriptions of ecological validity (see section 1. Introduction, in yellow).
2) Providing brief epidemiological context on the prevalence of MCI and AD in ageing populations would emphasise the urgency of developing new diagnostic tools.
Response 2: We have added a short paragraph providing global prevalence estimates of MCI and dementia to emphasize the urgency of developing new diagnostic tools (see section 1. Introduction, in yellow).
3) Mentioning how current guidelines or clinical practice incorporate (or fail to incorporate) spatial memory testing would position the review within a practical healthcare framework.
Response 3: In the introduction, we now include a description of the limited role of spatial memory assessments in current diagnostic guidelines.
4) Adding a note on how immersive technologies compare with other emerging digital tools (e.g. tablet-based cognitive tests and AI-driven diagnostics) would provide a fuller picture of the technological landscape.
Response 4: A note has been added in the introduction situating immersive technologies alongside other emerging digital approaches, highlighting their complementary role (AI-driven diagnostics).
5) 'Immersive technologies' are introduced early on (line 77) – it may be worth defining VR, iVR and MR briefly in one concise sentence so that non-specialist readers immediately understand the distinctions.
Response 5: We inserted clarifying definitions early in the introduction distinguishing VR, immersive VR (iVR), and MR for non-specialist readers.
6) The terms 'diagnostic sensitivity' and 'intervention efficacy' could be introduced with a brief example to make the text more accessible to a broader scientific audience.
Response 6: We have added practical examples to make these terms accessible (see section 1. Introduction, in yellow).
7) Please, consider separating 'assessment' and 'rehabilitation/training' into two distinct subsections within the introduction to improve clarity.
Response 7: The Introduction has been restructured to include two distinct subsections: Assessment and Rehabilitation/Training.
8) The 'Materials and Methods' section mentions the period from January 2020 to March 2025. Explaining why this range was chosen would strengthen the rationale. While limiting the search to English is common, this should be justified (e.g. accessibility or resources for translation).
Response 8: We clarified that the period was selected to capture the most recent advancements and that English was chosen due to limited resources for translation (see sections 2.1. and 2.2. in yellow).
9) Please, indicate whether studies including mixed-age populations were included, and whether data for participants aged 50 years or over could be extracted separately.
Response 9: We specified that mixed-age samples were included only when results for participants aged 50 years or over could be extracted separately (see section 2.2. in yellow).
10) The screening process states that two reviewers screened studies independently, with a third reviewer resolving conflicts. It would be useful to state whether a standardised data extraction form or protocol was used to ensure consistency.
Response 10: We added that a standardized data extraction form was developed by the review team to ensure consistency across studies (see section 2.2. in yellow). Additionally, we have included information about the registration of the protocol in the OSF (Registration: https://doi.org/10.17605/OSF.IO/K2UX3). Despite the absence of formal protocol, the methodological approach was delineated prior to the analysis and synthesis of the data. The registration of the protocol, available in the OSF repository, was completed after the data extraction phase had begun.
11) Figure 1 is referenced, but the text could briefly describe the numbers at each stage (e.g. duplicates removed, abstracts screened, full texts assessed, reasons for exclusion).
Response 11: The text now reports numbers of duplicates removed, abstracts screened, full texts assessed, and reasons for exclusion (see second paragraph of section 2.2. Study Selection, in yellow).
12) The section does not mention how the data were extracted (e.g. participant characteristics, interventions, outcomes). It does not describe the risk of bias or quality assessment of the included studies. Including this information would strengthen the methodological rigour of the review (e.g. by using the Cochrane Risk of Bias Tool, the Joanna Briggs Institute checklist, or a similar tool).
Response 12: A description of the data extraction process has been included, and it has been clarified that methodological quality was assessed using the JBI checklists specific to each study design. To ensure transparency, a supplementary table has been incorporated. This table provides a comprehensive summary of the type of study, the author and year of publication, the checklist items with compliance, and the final quality score (see new section 2.3. Quality Assessment and Supplementary Materials, Table S1, and new section 3.7).
13) In Section 3.1 ('Included studies'), please indicate the balance between cross-sectional, longitudinal and intervention-based studies. Since 'variety of geographic locations' is mentioned later, it would be helpful to specify here how global the sample is (e.g. the number of continents or countries represented). Mention briefly whether most studies used VR, AR or mixed modalities to frame the rest of the review. The section would be more complete with a short statement summarising the main inclusion and exclusion criteria for selecting these 42 studies.
Response 13: Section 3.1 has been expanded to specify the proportion of cross-sectional, longitudinal, and intervention studies, the use of VR modalities, geographic coverage, and summarize main inclusion/exclusion criteria (see section 3.1. in yellow).
14) In Section 3.2 'Sample characteristics of included studies', you could add the following: a) Mean age and standard deviation across studies; b) Proportion of participants with higher vs. lower education levels; c) Representation of ethnic or cultural backgrounds (if available). You could also quantify roughly how many studies or participants fell into each cognitive category (e.g. % MCI, % AD, % healthy controls). While the lack of reporting is noted, it would be useful to place a stronger emphasis on how this gap impacts generalisability. If data exist, indicate whether participant attrition or recruitment bias was common (e.g. recruitment from clinics vs community samples). Mention whether control and clinical groups were typically balanced in number or if disparities were common.
Response 14: We have substantially revised Section 3.2 ‘Sample characteristics of included studies’ to address the following aspects (see section 3.2 in yellow):
- a) We have included mean ages and standard deviations across studies where available, and noted the variability in age ranges and sample sizes.
- b) We now provide information on education levels, highlighting the predominance of highly educated participants and discussing how this may limit the generalizability of findings, particularly to more socioeconomically vulnerable populations.
- c) We added a section on the representation of ethnic and cultural backgrounds, emphasizing the notable underreporting of this data and the predominance of studies conducted in WEIRD countries.
Additionally, we have quantified the proportions of participants falling into different cognitive categories across the included studies (e.g., MCI, AD, healthy controls, and subjective cognitive decline).
We also address recruitment sources, noting whether participants were drawn from clinical settings or community samples, and we comment on the presence of group size imbalances between clinical and control groups in several studies.
Finally, we highlight the lack of systematic reporting on attrition or dropout rates.
15) The first sentence of Section 3.3 lists low-immersion desktop VR, high-immersion HMD VR, semi-immersive desktop VR and AR in quick succession. This could be reorganised into a clearer paragraph, starting with the least immersive and moving to the most immersive. Briefly define desktop VR, semi-immersive VR, fully immersive VR (HMD) and AR right at the start, so that the reader has a mental framework before reading the details.
Response 15: We thank the reviewer for this suggestion. Section 3.3 has been reorganised to introduce the technological platforms in a logical order, from the least to the most immersive. At the beginning of the section (in yellow), we now include a concise set of definitions for desktop VR, semi-immersive VR, fully immersive VR (HMD), and AR, providing the reader with a clear conceptual framework before presenting detailed examples.
16) Multiple sentences in the HMD section reiterate that these systems offer stereoscopic 3D and motion tracking, allowing for realistic body movement. This information could be condensed into a single clear sentence before moving on to examples.
Response 16: The description of HMD features has been condensed into a single sentence highlighting stereoscopic 3D and motion-tracking capabilities. The section now transitions directly to examples of empirical studies, improving readability and avoiding redundancy.
17) Currently, examples from Castegnaro, McCracken and Tuena are given in separate sentences. These could be grouped together under the heading 'Several studies have demonstrated...' and their key findings could then be listed in a compact comparative way.
Response 17: We have grouped the examples of Castegnaro et al., McCracken et al., and Tuena et al. under a common framing sentence (“Several studies have demonstrated...”), and then presented their findings in a comparative format (see section 3.3. in yellow).
18) While the HMD section is very detailed, the desktop/semi-immersive section could be slightly expanded with an extra example of a practical application outside the lab to balance the depth of coverage.
Response 18: We expanded the desktop/semi-immersive section by including an additional example of an applied study outside the laboratory context (Section 3.3. in yellow). This addition provides a more balanced depth of coverage across platforms.
19) The AR section is short and might benefit from some discussion of why it has not been more widely adopted (beyond hardware weight and field of view), for example cost, limitations in software development, or integration with clinical workflows.
Response 19: The AR section has been expanded to discuss additional barriers to adoption, including high development costs, software limitations, and challenges in integration into existing clinical workflows (see section 3.3. in yellow).
20) Using transition phrases such as 'In contrast' or 'By comparison' between immersive HMDs, semi-immersive systems and AR would make the reader's journey smoother. Currently, the transition from HMDs to desktop VR feels abrupt.
Response 20: We revised Section 3.3 to include transition phrases (e.g., “In contrast,” “By comparison”) when moving between subsections. This improves narrative flow and makes the progression from one platform to another more coherent for the reader.
21) After describing all platforms, conclude with a synthesis sentence that positions them along the immersion spectrum and links immersion level to research applicability. For example: 'Ultimately, the choice of technological platform involves balancing ecological validity, participant tolerance and research objectives. Higher immersion generally offers richer spatial engagement, but requires greater resources and technical support.'
Response 21: At the end of Section 3.3 (in yellow) we added a synthesis sentence highlighting how each platform falls along the immersion spectrum, and how higher immersion enhances ecological validity and spatial engagement but requires greater resources and technical support.
22) The first two sentences of Section 3.4 are strong, but somewhat dense. You could make them more engaging by starting with a direct statement about why immersive technologies are uniquely suited to assessing spatial cognition (e.g. 'Immersive technologies can precisely recreate real-world navigation challenges, allowing researchers to...').
Response 22: We revised the opening of Section 3.4 to begin with a direct, engaging statement (see section 3.4, in yellow).
23) The list of the four main task categories is good, but it would be even better if you briefly linked each one to its main cognitive demand in the same sentence, to give the reader a 'map' before the detailed descriptions.
Response 23: We have modified the text to briefly indicate the main cognitive demand of each task type (e.g., “Wayfinding tasks assess route learning and decision-making,” “Path integration tasks probe self-motion and spatial updating,” etc.). This gives readers a clear “map” before the detailed descriptions (see section 3.4, in yellow).
24) Currently, each task type starts abruptly ('Wayfinding tasks...', 'Thirteen studies...'). Consider adding linking phrases such as 'The most common were...', 'A second major category...', 'In contrast...', 'Finally...'. This would help to keep the reader oriented in the progression from the most to the least frequently studied task type.
Response 24: We have revised Section 3.4 so that each task category is introduced with linking phrases, improving readability and helping the reader follow the progression from the most to least common paradigms (see section 3.4, in yellow).
25) In the object-location memory section, the phrase 'hippocampal and entorhinal cortex functioning' could be simplified to 'hippocampus and entorhinal cortex' to avoid repetition of 'functioning'. Similarly, 'may not only assess but also support' could be tightened to 'may both assess and support'.
Response 25: We have simplified these expressions as suggested to avoid redundancy and improve conciseness (see section 3.4, in yellow).
26) Currently, the 3.4 section concludes with the egocentric vs. allocentric strategy example, which is interesting, but does not provide a summary of the four-task overview. Adding a final synthesis sentence could link the categories back to their diagnostic and rehabilitative potential, for example: 'Collectively, these spatial task paradigms tap into distinct yet complementary cognitive processes, providing sensitive markers of early neurodegenerative changes and potential avenues for targeted cognitive training.'
Response 26: We added a concluding synthesis sentence (see section 3.4, in yellow): “Collectively, these spatial task paradigms tap into distinct yet complementary cognitive processes, providing sensitive markers of early neurodegenerative changes and potential avenues for targeted cognitive training.”
27) The '3.5. Outcome Measures and Diagnostic Sensitivity' lists many measures, but does not explain why each category (behavioural, physiological and neuroimaging) is important in the context of VR cognitive diagnostics. It is suggested that a brief note is added to explain how these measures contribute to early detection or differential diagnosis (e.g. physiological measures detect subtle stress responses and neuroimaging links behaviour to neural substrates).
Response 27: We expanded Section 3.5 to clarify the diagnostic importance of each outcome type: behavioral measures capture observable navigation performance; physiological measures reveal subtle stress or cognitive load responses; and neuroimaging links behavioral outcomes to neural substrates, enhancing differential diagnosis (see section 3.5. in yellow).
28) Behavioural measures are clearly explained, but physiological and neuroimaging measures are only briefly mentioned. Please suggest adding at least one more concrete example of a physiological or EEG-based study to balance coverage.
Response 28: We incorporated additional examples of physiological and EEG-based studies (Amaefule et al., 2023 and Hanert et al., 2024) (see section 3.5).
29) The phrase 'several immersive tools demonstrated superior diagnostic sensitivity' could be expanded upon with a brief comparison — how much better were they compared to standard tests, if the data is available? If possible, please note the sensitivity or specificity percentages or effect sizes from the cited studies.
Response 29: We revised this section to provide quantitative comparisons where available. For example, Bayahya et al. (2021) reported 97.2% accuracy in detecting dementia using VR tasks compared to Mini-Cog, and Castegnaro et al. (2023) showed that angular errors in iVR path integration tasks distinguished MCI+ from MCI– with high effect sizes (see section 3.5. in yellow).
30) Enjoyment, safety and comfort are mentioned, but their role in diagnostic interpretation is not explained. Please, clarify whether these factors influence test performance, compliance or validity.
Response 30: We added a short clarification noting that enjoyment, safety, and comfort directly influence compliance, test validity, and long-term feasibility in older populations. For example, platforms that reduce cybersickness and enhance comfort increase participant retention and the reliability of repeated testing (see section 3.5. in yellow).
31) The section would be clearer if it were grouped into the following categories: (a) behavioural outcomes, (b) physiological outcomes, (c) neuroimaging outcomes and (d) multimodal approaches.
Response 31: We reorganized Section 3.5 into four clear categories: Behavioral outcomes, Physiological outcomes, Neuroimaging outcomes, and Multimodal approaches (see section 3.5. in yellow).
32) You could also consider mentioning ecological validity, i.e. how closely VR measures reflect real-world navigation and cognition, as this is an important reason for adopting these methods over traditional ones.
Response 32: We explicitly mention ecological validity in Section 3.5, noting that VR measures more closely replicate real-world navigation challenges than traditional tests, thereby enhancing their translational relevance to clinical practice.
33) The '3.6. Interventions and Training Effects' section states that most interventions were brief, but does not address whether any studies evaluated the longevity of training effects following the intervention period (i.e. follow-up testing). Including such data would clarify the potential for long-lasting cognitive benefits.
Response 33: We have expanded Section 3.6 to specify that none of the intervention studies included long-term follow-up testing with only one study including 3 months follow-up (see section 3.6. in yellow).
34) It would also be valuable to discuss whether improvements in spatial memory achieved through VR transfer to functional tasks in real-world navigation, daily living or independence, especially given that this is a key goal for ageing populations.
Response 34: Section 3.6 now includes discussion of transfer effects (see section 3.6. in yellow).
35) While EEG findings are mentioned, other neuroimaging or physiological markers (e.g. fMRI connectivity changes or neuroplasticity biomarkers) from intervention studies could be highlighted to demonstrate why VR might be effective.
Response 35: We expanded Section 3.6 to highlight the need for neuroimaging and physiological evidence (see section 3.6. in yellow).
36) It would also be useful to briefly note whether any intervention studies included diverse participant groups (e.g. those with varying cultural backgrounds, levels of digital literacy or comorbidities), as this affects generalisability.
Response 36: We have added a paragraph in Section 3.6. The studies recruited digitally literate older adults, including MCI patients, with relatively high education levels, which limits generalizability.
37) Including a sentence on practical barriers (e.g. cybersickness, equipment cost and accessibility for mobility-impaired older adults) would make the review more balanced and realistic.
Response 37: We agree and have expanded Section 3.6 to explicitly address practical barriers such as cybersickness, equipment costs, and challenges for participants with mobility impairments (see section 3.6. in yellow).
38) Some sentences, particularly those introducing VR's theoretical potential (lines 341–345), could be condensed slightly to avoid repetition with earlier sections describing VR's ecological validity.
Response 38: We revised Section 3.6 to condense sentences introducing VR’s theoretical potential, avoiding repetition.
39) The phrase 'may enhance encoding and retrieval processes' (lines 356–357) appears more than once in different forms; one instance could be reworded or omitted for concision.
Response 39: We agree and have reworded one of the repeated instances to avoid redundancy.
40) In the '3.7. Limitations across studies' section, please note that studies with positive or novel findings are more likely to be published, which could skew the literature towards optimistic outcomes. The scarcity of null or negative findings may overestimate the effectiveness of VR.
Response 40: We have added a statement in the new Section 3.8 acknowledging the potential for publication bias, noting that the scarcity of null or negative results may skew the literature towards optimistic conclusions and could lead to an overestimation of VR’s effectiveness (see section 3.8. in yellow).
41) Some studies may lack robust control groups or adequate blinding, which makes it difficult to distinguish VR-specific benefits from placebo, novelty or engagement effects. In older adults in particular, performance improvements may partly reflect familiarity with the hardware or software rather than genuine spatial memory enhancement. Few studies explicitly separate these effects.
Response 41: Section 3.8 (renumbered) now notes that the lack of robust control groups and blinding procedures limits the ability to disentangle genuine VR-specific effects from novelty, placebo, or engagement effects. We further highlight that in older adults, improvements may reflect increasing familiarity with the hardware/software, and that very few studies explicitly addressed this confound (see new section 3.8. in yellow).
42) While VR offers ecological realism, some highly realistic environments may introduce uncontrolled variability, which complicates the interpretation of cognitive outcomes. Cultural familiarity with environmental cues (e.g. building styles or street layouts) could influence navigation performance yet this is rarely considered in task design.
Response 42: We have expanded Section 3.8 to indicate that highly realistic VR environments may introduce uncontrolled variability, which complicates interpretation. We now also acknowledge that cultural familiarity with environmental cues such as street layouts or architectural styles may influence performance, but this factor is rarely considered in task design (see new section 3.8. in yellow).
43) Some details about input device types (e.g. joysticks, treadmills, teleportation and walking) could be condensed into a single sentence, as these specifics might be more suited to the methods review than the limitations section. The focus here should be on why the heterogeneity matters, rather than providing an exhaustive list of devices.
Response 43: We revised the limitations section to condense discussion of input devices into a single sentence, emphasizing instead why heterogeneity in interfaces matters (e.g., variability in immersion levels and motor demands may influence cognitive outcomes and comparability across studies) (see new section 3.8. in yellow).
44)The lines about cybersickness prevalence (397-399) could be tightened slightly without losing clarity by summarising the symptoms and triggers more concisely, for example.
Response 44: We have condensed the discussion of cybersickness to provide a concise summary of the most common symptoms (nausea, dizziness, disorientation) and typical triggers (visual-vestibular mismatch, prolonged exposure), without repeating details unnecessarily (see new section 3.8. in yellow).
45) There is no mention of how the data were synthesised, or whether this was qualitative, quantitative, or both. If a meta-analysis or effect size calculation was considered, this should be mentioned briefly. Even if only a narrative synthesis was performed, stating the approach would add clarity.
Response 45: We have clarified in the new Section 3.8 (last paragraph, in yellow) that a narrative synthesis approach was adopted due to the heterogeneity of study designs, outcomes, and populations. A narrative approach was chosen to highlight trends, methodological gaps, and pre-vailing trends.
46) The discussion of limitations and challenges (lines 102–106) could be expanded to include participant acceptability, accessibility for people with mobility impairments and the cultural adaptation of VR environments.
Response 46: We expanded the discussion of limitations to include participant acceptability, accessibility issues for those with mobility impairments, and the need for cultural adaptation of VR environments to ensure inclusivity and generalizability (see new section 3.8. in yellow).
47) In the 'Discussion' section, 'MC' should likely be 'MCI' (mild cognitive impairment) in lines 437-438 for consistency.
Response 47: We corrected this typographical error; “MC” has been replaced with “MCI” throughout the manuscript for consistency.
48) The review notes 'improved performance in real-world settings' after VR training, but does not clarify whether these improvements were sustained. This could be explicitly tied to the limitation regarding the lack of long-term follow-up.
Response 48: We revised the discussion to clarify that while some studies report improved performance in real-world navigation after VR training, few assessed whether these gains were sustained over time (see section 4. in yellow).
49) Although the discussion mentions small sample sizes and lack of diversity, it could highlight how these factors might bias the results (e.g. overestimation of VR efficacy in technologically literate participants).
Response 49: We agree and have added text explaining how small sample sizes and lack of diversity may bias results, for instance by overestimating efficacy in more technologically literate or motivated participants (see section 4. in yellow).
50) The statement on cybersickness could be improved with the inclusion of data, such as incidence rates or its effect on dropout rates.
Response 50: We strengthened this section by adding quantitative data where available (see section 4. in yellow).
51) While neurobiological validation is identified as scarce, the section could specify which types of validation are most urgently needed (e.g. establishing a link between specific VR navigation metrics and the integrity of the hippocampus and entorhinal cortex).
Response 51: We now specify that the most urgently needed validation involves linking VR-based navigation metrics to hippocampal and entorhinal cortex integrity, given their early involvement in MCI and AD (see section 4. in yellow).
52) The review mentions multimodal biomarkers (fMRI, PET, etc.) in its recommendations but does not discuss the practical or ethical challenges of integrating these in large-scale or at-home settings. Adding this information would strike a balance between feasibility and aspiration.
Response 52: We have added a note on practical and ethical challenges, including cost, accessibility, participant burden, and data privacy issues. We emphasize that while multimodal biomarkers are valuable, widespread clinical use will require pragmatic solutions (see section 4. in yellow).
53) While the ecological validity of VR is praised, the section could acknowledge that excessive realism or complexity could overwhelm older adults with cognitive decline, potentially impacting their test performance.
Response 53: We have revised the discussion to acknowledge that excessive realism or complexity in VR environments can overwhelm older adults with cognitive decline, potentially impacting their task performance. We now also emphasize that simplified and intuitive designs may reduce such cognitive overload and enhance usability (see section 4. in yellow, and last reference).
54) The final paragraph combines recommendations for research (e.g. longitudinal studies and diverse samples) with recommendations for clinical translation (e.g. standardised protocols and wearable sensors). Separating these into two distinct subthemes would improve clarity.
Response 54: We revised the final paragraph of the Discussion to separate recommendations into two subthemes: one addressing future research needs (e.g. longitudinal designs, diverse sampling, neurobiological validation) and one addressing clinical translation (e.g. standardised VR protocols, integration with wearable sensors) (see section 4. in yellow).
55) In the 'Conclusions' section, briefly reiterate how VR assessments could transform early diagnosis and intervention strategies for MCI and AD within real-world healthcare systems.
Response 55: We revised the Conclusions to reiterate that VR assessments have strong potential to transform early diagnosis and intervention for MCI and AD by providing ecologically valid, scalable tools for use in real-world healthcare systems (see section 5. in yellow).
56) While the 'Discussion' section acknowledges the role of VR in training and rehabilitation as well as assessment, this is absent from the conclusions.
Response 56: We added a sentence in the Conclusions noting that immersive technologies may serve both as sensitive diagnostic tools and as platforms for cognitive training and rehabilitation (see section 5. in yellow).
57) Including a brief reminder that small sample sizes, a lack of longitudinal data and cybersickness are still barriers would make the conclusion more balanced.
Response 57: We included a sentence in the Conclusions reminding readers that small sample sizes, limited longitudinal data, and cybersickness remain key barriers to clinical adoption (see section 5. in yellow).
58) Including one sentence calling for the standardisation of VR protocols and the inclusion of more diverse participant groups would mirror the strong recommendations in the discussion.
Response 58: We added a final sentence in the Conclusions explicitly calling for protocol standardization and recruitment of more diverse participant groups in future studies (see section 5. in yellow).